# Preserve-Then-Quantize:
# Balancing Rank Budgets for Quantization Error Reconstruction in LLMs

Yoonjun Cho [* 1]   Dongjae Jeon [* 1]   Soeun Kim [2]   Moongyu Jeon [2]   Albert No [2]

## Abstract

Quantization Error Reconstruction (QER) reduces accuracy loss in Post-Training Quantization (PTQ) by approximating weights as $\mathbf{W} \approx \mathbf{Q} + \mathbf{LR}$, using a rank-$r$ correction to reconstruct quantization error. Prior methods devote the full rank budget to error reconstruction, which is suboptimal when $\mathbf{W}$ has intrinsic low-rank structure and quantization corrupts dominant directions. We propose Structured Residual Reconstruction (SRR), a rank-allocation framework that preserves the top-$k$ singular subspace of the activation-scaled weight before quantization, quantizes only the residual, and uses the remaining rank $r - k$ for error reconstruction. We derive a theory-guided criterion for selecting $k$ by balancing quantization-exposed energy and unrecoverable error under rank constraints. We further show that resulting $\mathbf{Q} + \mathbf{LR}$ parameterization naturally supports Quantized Parameter-Efficient Fine-Tuning (QPEFT), and stabilizes fine-tuning via gradient scaling along preserved directions. Experiments demonstrate consistent perplexity reductions across diverse models and quantization settings in PTQ, along with a 5.9 percentage-point average gain on GLUE under 2-bit QPEFT. The project page is available at https://ai-isl.github.io/srr.

## 1. Introduction

Post-training quantization (PTQ) is a primary approach for reducing the memory footprint and inference cost of large language models (LLMs) by converting full-precision weights to low-bit representations (Nagel et al., 2021; Frantar et al., 2023). However, aggressive low-bit PTQ often

causes substantial accuracy degradation, especially in the 3/2-bit regime (Yao et al., 2022; Lin et al., 2024; Tseng et al., 2024). A widely adopted remedy is *quantization error reconstruction* (QER), which approximates the full-precision weight matrix as $\mathbf{W} \approx \mathbf{Q} + \mathbf{LR}$, where $\mathbf{Q} = \mathcal{Q}(\mathbf{W})$ is produced by a quantizer $\mathcal{Q}(\cdot)$ and $\mathbf{LR}$ (rank $\leq r$) is constructed to recover information lost during quantization (Yao et al., 2024). Modern QER methods further incorporate activation statistics by fitting the correction in a scaled space defined by a matrix $\mathbf{S}$, improving alignment with input-dependent layer behavior (Liu et al., 2024; Zhang et al., 2024a; 2025). In practice, $\mathbf{LR}$ is typically obtained via truncated SVD.

Despite their success, existing QER methods make an implicit allocation decision: given a rank budget $r$, they devote *all* rank capacity to approximating the (scaled) quantization residual $\mathbf{S}(\mathbf{W} - \mathbf{Q})$. This choice is appropriate only when the residual itself has low effective rank. In low-bit regimes, quantization error is often dense and high-rank, and residual-only reconstruction can become inefficient. More importantly, this pipeline ignores a structural fact about transformer weights: in the activation-scaled space, $\mathbf{SW}$ is often highly anisotropic, with most energy concentrated in a small set of dominant singular directions (Yuan et al., 2023b; Wang et al., 2025). Quantizing these dominant directions injects disproportionately large scaled error, forcing a rank-limited correction to spend capacity repairing avoidable distortion in the most informative subspace. This suggests that the low-rank budget should not serve a single purpose. Instead, part of the rank capacity can be used to *preserve* the dominant low-rank structure of $\mathbf{SW}$ before quantization, while the remaining capacity is reserved to *reconstruct* the quantization error introduced on the residual. The relative importance of preservation versus reconstruction depends on the layer and scaling statistics, making rank allocation a layer- and matrix-specific decision. These observations lead to a central question: *under a fixed rank budget, how should we balance preservation of dominant structure and reconstruction of quantization error?*

We propose **Structured Residual Reconstruction (SRR)**, a *preserve-then-quantize* framework that makes this balance explicit. SRR allocates $k$ ranks to preserve the dominant subspace of $\mathbf{SW}$ (bypassing quantization on these directions),

---

[1]Department of Computer Science, Yonsei University [2]Department of Artificial Intelligence, Yonsei University. Correspondence to: Albert No <albertno@yonsei.ac.kr>.

*Proceedings of the 43rd International Conference on Machine Learning*, Seoul, South Korea. PMLR 306, 2026. Copyright 2026 by the author(s).

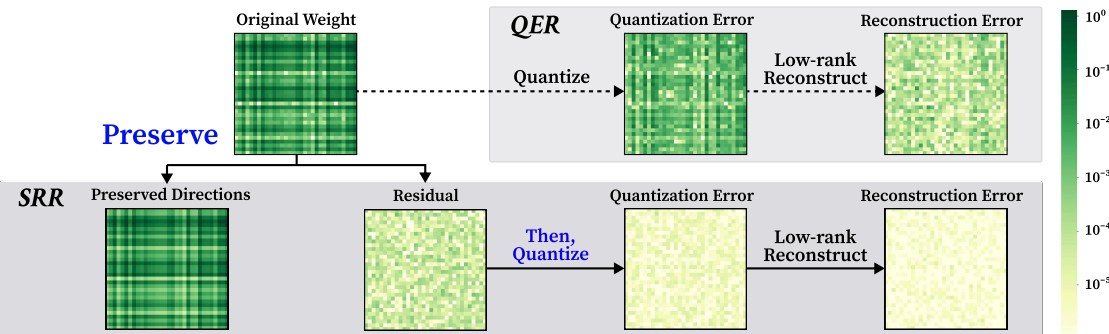

*Figure 1. Preserve-then-quantize* mechanism in Structured Residual Reconstruction (SRR), compared with standard QER. **Top**: In QER, quantizing the entire weight matrix destroys the intrinsic low-rank structure of the original weight, leaving errors that cannot be recovered by low-rank reconstruction. **Bottom**: In contrast, SRR preserves the dominant low-rank structure prior to quantization, preventing structural damage and resulting in a substantially smaller reconstruction error $\|\mathbf{W} - \mathbf{Q} - \mathbf{LR}\|_F$ under the same rank budget.

quantizes only the remaining residual, and uses the remaining $r - k$ ranks to reconstruct the induced quantization error. Under the same rank budget, SRR prevents quantization from corrupting high-energy directions and uses the low-rank capacity where it is most effective. A key component of SRR is selecting the split point $k$. SRR derives a simple, theory-guided rule that balances (i) allocating rank $k$ to preserve dominant structure before quantization and (ii) allocating the remaining rank $r - k$ to reconstruct the quantization error. To make it practical, SRR uses a lightweight one-shot random-matrix probe as a proxy for the normalized quantization-error spectrum, enabling fast and robust layer-wise selection of $k$ without expensive enumeration.

SRR also extends naturally to Quantized Parameter-Efficient Fine-Tuning (QPEFT) (Hu et al., 2022; Li et al., 2024; Guo et al., 2024; Zhang et al., 2025), where the quantized backbone is frozen and only the low-rank component is trained. SRR provides a strong initialization for the adapter through its decomposition of $\mathbf{W} \approx \mathbf{Q} + \mathbf{LR}$, while the rank-allocation clarifies the roles of different low-rank directions: preserved directions should remain stable, and reconstructed directions provide adaptation capacity. Accordingly, we stabilize QPEFT by attenuating updates along the preserved subspace while allowing the remaining directions to adapt.

Across extensive experiments in both weight-only PTQ and QPEFT, SRR consistently outperforms prior QER methods under identical rank budgets, demonstrating that *balancing* preservation and reconstruction is critical for accurate low-bit approximation of $\mathbf{W} \approx \mathbf{Q} + \mathbf{LR}$.

## 2. Preliminaries

**Quantization Error Reconstruction (QER).** Post-Training Quantization (PTQ) compresses a pretrained model by mapping full-precision weights to low-bit representations (Nagel et al., 2020), often degrading accuracy in low-bit regimes. QER mitigates this loss by quantizing first and

then adding a low-rank correction to compensate for the quantization error (Yao et al., 2024; Zhang et al., 2024a; Liu et al., 2024; Zhang et al., 2025; Lee et al., 2025).

Consider a linear layer with output $\mathbf{y} = \mathbf{xW}$, where $\mathbf{W} \in \mathbb{R}^{m \times n}$ and $\mathbf{x} \in \mathbb{R}^m$. Let $\mathbf{Q} = \mathcal{Q}(\mathbf{W})$ denote the quantized weight, where $\mathcal{Q}(\cdot)$ is a (possibly activation-aware) quantizer. QER approximates $\mathbf{W}$ by

$$\mathbf{W} \approx \mathbf{Q} + \mathbf{LR}, \quad \mathbf{L} \in \mathbb{R}^{m \times r}, \ \mathbf{R} \in \mathbb{R}^{r \times n}, \ r \ll \min(m, n),$$

so that the quantized layer output becomes $\mathbf{y}_q = \mathbf{x}(\mathbf{Q} + \mathbf{LR}) \approx \mathbf{xW}$. Typically, in the *fine-tuning-free* setting that we focus on, $\mathbf{LR}$ is computed once from model weights.

A construction fits the correction to the residual $\mathbf{W} - \mathbf{Q}$ via the best rank-$r$ approximation in Frobenius norm:

$$\mathbf{LR} = \operatorname*{arg\,min}_{\operatorname{rank}(\boldsymbol{\Delta}) \leq r} \|\mathbf{W} - \mathbf{Q} - \boldsymbol{\Delta}\|_F = \mathrm{SVD}_r(\mathbf{W} - \mathbf{Q}),$$

as in ZeroQuant-V2 (Yao et al., 2024). While effective at reconstructing weights, this objective is agnostic to the input distribution and may not optimally preserve layer behavior.

**Activation-aware QER.** Recent QER methods incorporate activation statistics by reconstructing the residual in a scaled space (Liu et al., 2024; Zhang et al., 2024a; 2025). Let $\mathbf{S} \in \mathbb{R}^{m \times m}$ be an invertible scaling matrix derived from calibration activations. Activation-aware QER solves

$$\mathbf{LR} = \operatorname*{arg\,min}_{\operatorname{rank}(\boldsymbol{\Delta}) \leq r} \|\mathbf{S}(\mathbf{W} - \mathbf{Q} - \boldsymbol{\Delta})\|_F$$
$$= \mathbf{S}^{-1} \mathrm{SVD}_r(\mathbf{S}(\mathbf{W} - \mathbf{Q})), \qquad (1)$$

which emphasizes input directions that matter most for the layer output. Different choices of $\mathbf{S}$ recover existing activation-aware QER variants, including heuristic scalings derived from calibration data, such as LQER and QERA-approx (Zhang et al., 2024a; 2025), and exact solutions

based on Fisher information or second-order approximations, including QERA-exact and related approaches (Liu et al., 2024; Zhang et al., 2025; Saha et al., 2024; Cho et al., 2025), which better capture output-sensitive directions.

**Quantized Parameter-Efficient Fine-Tuning (QPEFT).** PEFT methods such as LoRA (Hu et al., 2022) adapt a pre-trained model by learning a low-rank update while keeping the base weights frozen. QPEFT extends this to quantized backbones by freezing $\mathbf{Q}$ and training a low-rank adapter, typically of the form $\mathbf{W} \approx \mathbf{Q} + \mathbf{LR}$, to recover task performance (Dettmers et al., 2023). Because quantization introduces a non-negligible mismatch between $\mathbf{W}$ and $\mathbf{Q}$, several QPEFT methods improve stability by initializing the adapter to approximate the quantization residual before fine-tuning (Li et al., 2024; Guo et al., 2024; Zhang et al., 2025). For example, LoftQ and LQ-LoRA refine such initializations with iterative quantization/reconstruction procedures, while QERA (Zhang et al., 2025) directly adopts a one-shot QER formulation for both PTQ and QPEFT initialization.

# 3. Motivation: Rank Allocation for Quantization Error Reconstruction

Activation-aware QER methods (Liu et al., 2024; Zhang et al., 2024a; 2025) improve low-bit PTQ by quantizing $\mathbf{W}$ to $\mathbf{Q} = \mathcal{Q}(\mathbf{W})$ and then adding a rank-$r$ correction to fit the scaled residual (cf. (1)). This pipeline assumes that the scaled quantization error $\mathbf{S}(\mathbf{W} - \mathbf{Q})$ is effectively low-rank, so a rank-$r$ approximation can remove most distortion.

However, in the low-bit regime, this assumption often fails. Entrywise quantization introduces a dense perturbation, and after activation scaling the error typically becomes high-rank. Importantly, high-rank quantization error can arise even when the scaled weight $\mathbf{SW}$ itself exhibits strong low-rank structure. As a simple limiting example, if $\mathrm{rank}(\mathbf{SW}) \leq r$, then the layer can be represented exactly with rank-$r$ capacity in the scaled space, yet naive QER can still leave nonzero reconstruction error because $\mathbf{S}(\mathbf{W} - \mathcal{Q}(\mathbf{W}))$ is generally not rank-$r$. This highlights a mismatch between what the rank budget could represent and what the standard pipeline forces it to correct.

The root cause is structural. Transformer weights in the scaled space $\mathbf{SW}$ are typically highly anisotropic, with a small number of leading singular directions carrying a large fraction of the energy (Yuan et al., 2023b; Wang et al., 2025). Quantizing these dominant directions injects disproportionately large scaled error, and a rank-limited correction must spend capacity repairing avoidable distortion in the most informative subspace. This motivates treating the rank budget as a resource to be explicitly split between two roles: (i) preserving dominant structure *before* quantization and (ii) reconstructing the remaining quantization error.

We capture this idea with an explicit rank-allocation formulation. For $k \in \{0, \ldots, r\}$, we preserve a rank-$k$ component $\Delta_1$, quantize the residual $\mathbf{W} - \Delta_1$, and use the remaining rank $r - k$ term $\Delta_2$ to reconstruct the induced error:

$$\min_{0 \leq k \leq r} \min_{\substack{\mathrm{rank}(\Delta_1) \leq k \\ \mathrm{rank}(\Delta_2) \leq r-k}} \left\| \mathbf{S}\big( \mathbf{W} - (\Delta_1 + \mathcal{Q}(\mathbf{W} - \Delta_1) + \Delta_2) \big) \right\|_F. \tag{2}$$

Here, QER corresponds to $k = 0$, where all rank is devoted to residual correction. At the other extreme, $k = r$ mirrors LQ-LoRA (Guo et al., 2024) and SVDQuant (Li et al., 2025), which prioritize preserving low-rank structure over residual reconstruction. More broadly, (2) exposes the central question: how should a fixed rank budget be divided between preserving structure and reconstructing quantization error?

In practice, the answer is matrix-dependent. Increasing $k$ reduces the energy exposed to quantization but leaves fewer ranks $(r - k)$ to reconstruct the remaining error; decreasing $k$ does the opposite. The optimal $k$ therefore depends on the spectral decay of $\mathbf{SW}$ and the geometry induced by $\mathbf{S}$. Figure 2 shows this is not merely theoretical: even within the same model, different projection matrices achieve their minimum reconstruction error at different $k$ values. This motivates a method that selects $k$ in a layer- and matrix-specific manner and realizes the rank allocation in (2), which we develop next as Structured Residual Reconstruction (SRR).

# 4. Structured Residual Reconstruction (SRR)

We implement a rank-allocation view in a *plug-and-play*, *fine-tuning-free* manner. Given a scaling matrix $\mathbf{S}$ (from activation statistics), a quantizer $\mathcal{Q}(\cdot)$, and a total rank budget $r$, we allocate $k$ ranks to preserve the dominant subspace of $\mathbf{SW}$ and the remaining $r - k$ ranks to reconstruct the quantization error. This explicitly trades off between subspace preservation and error correction under a fixed rank budget.

### 4.1. Preserve-Quantize-Reconstruct with a Rank Split

Fix a split $k \in \{0, \ldots, r\}$. Structured Residual Reconstruction (SRR) first extracts the dominant rank-$k$ directions in the scaled weight space and maps it back to the original weight space:

$$\mathbf{L}_k^{(1)} \mathbf{R}_k^{(1)} := \mathbf{S}^{-1} \, \mathrm{SVD}_k (\mathbf{SW}) .$$

It then quantizes the remaining component,

$$\mathbf{Q}_k := \mathcal{Q}\big( \mathbf{W} - \mathbf{L}_k^{(1)} \mathbf{R}_k^{(1)} \big),$$

and defines the corresponding quantization error

$$\mathbf{E}_k := \mathbf{W} - \mathbf{L}_k^{(1)} \mathbf{R}_k^{(1)} - \mathbf{Q}_k.$$

Finally, SRR uses the remaining rank budget $r - k$ to reconstruct the scaled error via truncated SVD:

$$\mathbf{L}_k^{(2)} \mathbf{R}_k^{(2)} := \mathbf{S}^{-1} \, \mathrm{SVD}_{r-k} (\mathbf{SE}_k) .$$

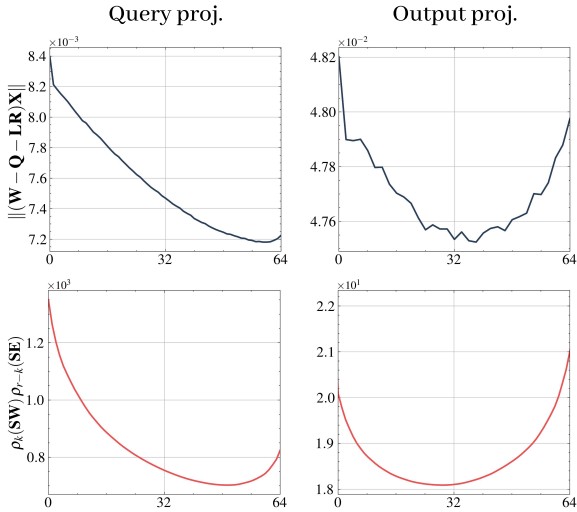

*Figure 2.* Alignment between reconstruction error and the rank-selection objective. Reconstruction error (**Top**) and surrogate objective (**Bottom**) as functions of the preserved rank $k$ under a rank budget of $r = 64$. Similar trends across $k$ support the use of the objective for selecting $k^\star$. Results are shown for the Query (*Left*) and Output (*Right*) projections in LLaMA-2 7B, layer 10. Additional results for other projections are provided in Appendix B.3.

The resulting approximation is

$$\widehat{\mathbf{W}}_{\mathrm{SRR}}(k) := \mathbf{L}_k^{(1)}\mathbf{R}_k^{(1)} + \mathbf{Q}_k + \mathbf{L}_k^{(2)}\mathbf{R}_k^{(2)}.$$

SRR produces two low-rank terms, consisting of a rank-$k$ component that preserves the dominant subspace and a rank-$(r - k)$ component that reconstructs the residual:

$$\mathbf{L}_k = [\mathbf{L}_k^{(1)}, \mathbf{L}_k^{(2)}], \qquad \mathbf{R}_k = \begin{bmatrix} \mathbf{R}_k^{(1)} \\ \mathbf{R}_k^{(2)} \end{bmatrix},$$

so $\mathbf{L}_k\mathbf{R}_k = \mathbf{L}_k^{(1)}\mathbf{R}_k^{(1)} + \mathbf{L}_k^{(2)}\mathbf{R}_k^{(2)}$ has rank at most $r$. Thus, at inference time, SRR takes the form $\widehat{\mathbf{W}} = \mathbf{Q}_k + \mathbf{L}_k\mathbf{R}_k$.

### 4.2. Selecting $k$ by Modeling Reconstruction Error

SRR chooses the split $k$ to minimize the scaled reconstruction error, and then spends $r - k$ ranks on correcting quantization noise. For a fixed $k$, SRR quantizes $\mathbf{W} - \mathbf{L}_k^{(1)}\mathbf{R}_k^{(1)}$ and induces the quantization error

$$\begin{aligned} \mathbf{E}_k &:= \mathrm{E}_{\mathcal{Q}}\left(\mathbf{W} - \mathbf{L}_k^{(1)}\mathbf{R}_k^{(1)}\right) \\ &= \mathbf{W} - \mathbf{L}_k^{(1)}\mathbf{R}_k^{(1)} - \mathcal{Q}\left(\mathbf{W} - \mathbf{L}_k^{(1)}\mathbf{R}_k^{(1)}\right), \end{aligned}$$

where $\mathrm{E}_{\mathcal{Q}}(\cdot)$ denotes the quantization error corresponding to the quantizer $\mathcal{Q}$. Since $\mathbf{S}\mathbf{L}_k^{(2)}\mathbf{R}_k^{(2)}$ is the best rank-$(r - k)$ approximation of $\mathbf{S}\mathbf{E}_k$, the optimal scaled error for $k$ is

$$\begin{aligned} \mathcal{L}(k) &:= \|\mathbf{S}(\mathbf{W} - \widehat{\mathbf{W}}_{\mathrm{SRR}}(k))\|_F \\ &= \|\mathbf{S}\mathbf{E}_k - \mathrm{SVD}_{r-k}(\mathbf{S}\mathbf{E}_k)\|_F. \end{aligned}$$

Define the rank-$p$ unrecoverable energy ratio of matrix $\mathbf{A}$ as

$$\rho_p(\mathbf{A}) := \frac{\|\mathbf{A} - \mathrm{SVD}_p(\mathbf{A})\|_F^2}{\|\mathbf{A}\|_F^2} = 1 - \frac{\sum_{j=1}^p \sigma_j(\mathbf{A})^2}{\|\mathbf{A}\|_F^2}.$$

By the optimality of truncated SVD, the optimal scaled reconstruction error for a given split $k$ admits the factorization

$$\mathcal{L}(k)^2 = \|\mathbf{S}\mathbf{E}_k\|_F^2\, \rho_{r-k}(\mathbf{S}\mathbf{E}_k). \qquad (3)$$

Equation (3) separates the problem into two terms: (i) the *scale* of the scaled quantization error $\|\mathbf{S}\mathbf{E}_k\|_F$ and (ii) the *spectral concentration* term $\rho_{r-k}(\mathbf{S}\mathbf{E}_k)$, which measures how much of that error remains after the best rank-$(r - k)$ reconstruction. Directly minimizing (3) over $k$ is expensive because $\mathbf{E}_k$ depends on $k$, and evaluating each candidate requires an additional quantization and an SVD of $\mathbf{S}\mathbf{E}_k$. SRR models these two terms with assumptions that depend primarily on the quantizer $\mathcal{Q}$ and bitwidth, not on $k$.

The first assumption follows the standard additive quantization noise view of rounding-based (uniform) quantization:

**Assumption 4.1** (quantization error has approximately constant relative scale). For a fixed quantizer $\mathcal{Q}$ and bitwidth, the scaled quantization error energy is approximately proportional to the scaled input energy:

$$\|\mathbf{S}\, \mathrm{E}_{\mathcal{Q}}(\mathbf{A})\|_F \approx \eta_{\mathcal{Q}}\|\mathbf{S}\mathbf{A}\|_F,$$

where $\eta_{\mathcal{Q}}$ depends primarily on the quantization resolution and calibration (e.g., step sizes / clipping), and varies weakly across matrices $\mathbf{A}$ within a layer.

This assumption is justified in that, under a fixed quantizer configuration, quantization error remains bounded and its energy is proportional to the signal magnitude via the quantization step size, resulting in an approximately constant relative error rate for a fixed bitwidth (Alistarh et al., 2017).

Applying Assumption 4.1 to $\mathbf{A} = \mathbf{W} - \mathbf{L}_k^{(1)}\mathbf{R}_k^{(1)}$ gives

$$\begin{aligned} \|\mathbf{S}\mathbf{E}_k\|_F^2 &\approx \eta_{\mathcal{Q}}^2\|\mathbf{S}(\mathbf{W} - \mathbf{L}_k^{(1)}\mathbf{R}_k^{(1)})\|_F^2 \\ &= \eta_{\mathcal{Q}}^2\|\mathbf{S}\mathbf{W} - \mathrm{SVD}_k(\mathbf{S}\mathbf{W})\|_F^2 \\ &= \eta_{\mathcal{Q}}^2\rho_k(\mathbf{S}\mathbf{W})\,\|\mathbf{S}\mathbf{W}\|_F^2. \qquad (4) \end{aligned}$$

Thus, the scale term in (3) can be computed exactly from the spectrum of $\mathbf{S}\mathbf{W}$.

The next assumption captures the empirical observation that the normalized quantization residual behaves like unstructured noise after rounding.

**Assumption 4.2** (spectral proxy for quantization error). Let $\mathbf{E}$ be a random matrix with i.i.d. entries drawn from $\mathcal{U}[-1, 1]$. After normalization, the scaled quantization error exhibits a $k$-insensitive spectral profile, and we approximate

$$\rho_{r-k}(\mathbf{S}\mathbf{E}_k) \approx \rho_{r-k}(\mathbf{S}\mathbf{E}).$$

---

**Algorithm 1** Structured Residual Reconstruction (SRR)

---

**Require:** Weight $\mathbf{W} \in \mathbb{R}^{m \times n}$, scaling $\mathbf{S} \in \mathbb{R}^{m \times m}$, quantizer $\mathcal{Q}(\cdot)$, rank budget $r$

**Ensure:** $\mathbf{Q}, \mathbf{L}, \mathbf{R}$ such that $\widehat{\mathbf{W}} = \mathbf{Q} + \mathbf{L}\mathbf{R}$

  1: Sample $\mathbf{E} \in \mathbb{R}^{m \times n}$ with $\mathbf{E}_{ij} \sim \mathcal{U}[-1, 1]$
  2: $k^{\star} \leftarrow \arg\min_{0 \leq k \leq r} \rho_k(\mathbf{SW})\,\rho_{r-k}(\mathbf{SE})$      (Eq. (5))
  3: $\mathbf{L}^{(1)}\mathbf{R}^{(1)} \leftarrow \mathbf{S}^{-1}\,\mathrm{SVD}_{k^{\star}}(\mathbf{SW})$      (preserve)
  4: $\mathbf{Q} \leftarrow \mathcal{Q}(\mathbf{W} - \mathbf{L}^{(1)}\mathbf{R}^{(1)})$      (quantize)
  5: $\mathbf{E} \leftarrow \mathbf{W} - \mathbf{L}^{(1)}\mathbf{R}^{(1)} - \mathbf{Q}$      (quantization error)
  6: $\mathbf{L}^{(2)}\mathbf{R}^{(2)} \leftarrow \mathbf{S}^{-1}\,\mathrm{SVD}_{r-k^{\star}}(\mathbf{SE})$      (reconstruct)
  7: $\mathbf{L} \leftarrow [\mathbf{L}^{(1)}, \mathbf{L}^{(2)}], \quad \mathbf{R} \leftarrow \begin{bmatrix} \mathbf{R}^{(1)} \\ \mathbf{R}^{(2)} \end{bmatrix}$
  8: **return** $\mathbf{Q}, \mathbf{L}, \mathbf{R}$

---

This assumption follows from the quantization-noise model, which treats quantization error as approximately signal-independent random noise after normalization (Marco & Neuhoff, 2005; Lin et al., 2016; Meller et al., 2019).

Combining (3) with Assumptions 4.1–4.2 and (4) yields

$$\mathcal{L}(k)^2 \approx \eta_{\mathcal{Q}}^2 \, \|\mathbf{SW}\|_F^2 \, \rho_k(\mathbf{SW})\,\rho_{r-k}(\mathbf{SE}).$$

Since $\eta_{\mathcal{Q}}$ and $\|\mathbf{SW}\|_F$ are constant across $k$, SRR selects

$$k^{\star} = \arg\min_{0 \leq k \leq r} \rho_k(\mathbf{SW})\,\rho_{r-k}(\mathbf{SE}). \tag{5}$$

Evaluating (5) requires singular values of $\mathbf{SW}$ and of a single random probe $\mathbf{SE}$, enabling an efficient layer-wise search over $k \in \{0, \ldots, r\}$ without enumerating $\mathbf{E}_k$.

### 4.3. SRR Algorithm

With the split point $k^{\star}$ determined, SRR reduces to a four-step procedure under a fixed rank budget $r$. For each layer, SRR (i) selects the split $k^{\star}$, (ii) extracts a rank-$k^{\star}$ component from $\mathbf{SW}$, (iii) quantizes the remaining part, and (iv) reconstructs induced quantization error using the remaining $r - k^{\star}$ ranks. Algorithm 1 summarizes the full pipeline.

Note that SRR estimates $\rho_{r-k}(\mathbf{SE})$ using a single random draw (one probe) and reuses it for all $k$ and for the entire layer. At first glance, this may appear unstable because the proxy matrix is random. However, in practice it is highly stable at dimensions of transformer layer: the singular-value energy profile of $\mathbf{SE}$ concentrates, so $\rho_{r-k}(\mathbf{SE})$ varies very little between draws. Empirically, repeating the procedure with different random probes changes the selected $k^{\star}$ only marginally (typically within $\pm 1$), as shown in Appendix B.1. Consequently, this makes the one-shot probe a reliable and inexpensive substitute for enumerating $\mathbf{E}_k$.

Additionally, Line 6 in Algorithm 1 can be replaced by a single rank-$r$ reconstruction of the residual $\mathbf{W} - \mathbf{Q}$:

$$\mathbf{L}\mathbf{R} \leftarrow \mathbf{S}^{-1}\,\mathrm{SVD}_r(\mathbf{S}(\mathbf{W} - \mathbf{Q})). \tag{6}$$

This removes explicit $\mathrm{SVD}_{r-k^{\star}}(\cdot)$ while maintaining the same $k^{\star}$-dependent quantization step (Step (iv)). For fixed $\mathbf{Q}$, (6) is the best rank-$r$ correction in the scaled Frobenius norm by the Eckart-Young theorem (Eckart & Young, 1936), and empirically behaves like the intended split: leading components (rank $k^{\star}$) recover the preserved structure, while remaining ones (rank $(r - k^{\star})$) suppress quantization noise.

### 4.4. Application to QPEFT: Two-Component Adapters and Decoupled Updates

SRR extends directly to Quantized Parameter-Efficient Fine-Tuning (QPEFT), where the quantized weight is frozen and only a low-rank adapter is trained. Starting from the SRR solution at the selected split $k^{\star}$ (Algorithm 1), we fix the quantized backbone to the final output $\mathbf{Q}$. For notational simplicity, we denote $(\mathbf{L}^{(1)}, \mathbf{R}^{(1)}) := (\mathbf{L}_{k^{\star}}^{(1)}, \mathbf{R}_{k^{\star}}^{(1)})$ and $(\mathbf{L}^{(2)}, \mathbf{R}^{(2)}) := (\mathbf{L}_{k^{\star}}^{(2)}, \mathbf{R}_{k^{\star}}^{(2)})$. The trainable adapter is initialized as $\mathbf{L}\mathbf{R} := \mathbf{L}^{(1)}\mathbf{R}^{(1)} + \mathbf{L}^{(2)}\mathbf{R}^{(2)}$, and the resulting QPEFT weight during training is $\widehat{\mathbf{W}} = \mathbf{Q} + \mathbf{L}\mathbf{R}$. This initialization yields a high-fidelity approximation of the pre-trained weight, providing a stable starting point for QPEFT.

The two low-rank components play different roles: the factors $(\mathbf{L}^{(1)}, \mathbf{R}^{(1)})$ preserve the dominant subspace of $\mathbf{SW}$, while $(\mathbf{L}^{(2)}, \mathbf{R}^{(2)})$ target quantization-induced errors. As a result, the magnitudes of the induced low-rank components differ substantially: the singular values associated with $\mathbf{L}^{(1)}\mathbf{R}^{(1)}$ are much higher than those associated with $\mathbf{L}^{(2)}\mathbf{R}^{(2)}$ (Figure 3a). If a single update magnitude is applied to all factors, optimization may over-update the dominant directions or, conversely, under-utilize the residual directions. Either effect degrades training stability and can wash out the structural benefit of SRR initialization.

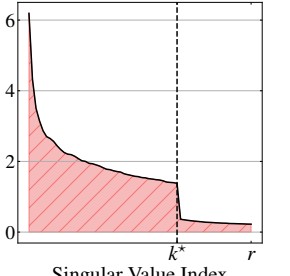 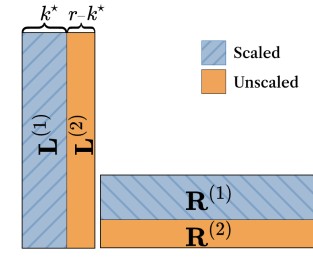

*(a)* Singular-value spectrum           *(b)* Illustration of low-rank term

*Figure 3.* (a) Singular-value spectrum with the selected split $k^{\star}$. (b) Illustration of the corresponding low-rank factors, where gradients along the $k^{\star}$ preserved directions are scaled, while the remaining $r - k^{\star}$ residual directions are left unscaled.

To account for scale separation, we regulate updates on the preserved directions via gradient scaling (Figure 3b). A simple and effective strategy is to attenuate its gradients:

$$\nabla_{\mathbf{L}^{(1)}\mathbf{R}^{(1)}}\mathcal{L} \leftarrow \gamma\,\nabla_{\mathbf{L}^{(1)}\mathbf{R}^{(1)}}\mathcal{L}, \qquad \gamma \in (0, 1), \tag{7}$$

| | Method | TinyLLaMA 1.1B | | Gemma-2 2B | | LLaMA-2 7B | | LLaMA-2 13B | | LLaMA-3.1 8B | | LLaMA-3.1 70B | |
|---|---|---|---|---|---|---|---|---|---|---|---|---|---|
| | | $r=32$ | $r=64$ | $r=32$ | $r=64$ | $r=32$ | $r=64$ | $r=32$ | $r=64$ | $r=32$ | $r=64$ | $r=32$ | $r=64$ |
| | BF16 | 13.98 | | 13.08 | | 8.71 | | 7.68 | | 7.55 | | 3.06 | |
| | *w-only* | 32.82 | | 41.13 | | 13.33 | | 10.25 | | 18.96 | | 16.46 | |
| | LQER | 21.95 | 20.63 | 22.99 | 21.37 | 14.51 | 15.14 | 9.18 | 9.13 | 12.39 | 11.90 | 6.85 | 6.69 |
| | **w/ SRR** | $21.03_{\pm0.05}$ | $20.22_{\pm0.09}$ | $21.67_{\pm0.17}$ | $20.92_{\pm0.17}$ | $11.22_{\pm0.09}$ | $11.03_{\pm0.01}$ | $9.09_{\pm0.00}$ | $8.97_{\pm0.00}$ | $12.37_{\pm0.03}$ | $11.61_{\pm0.03}$ | $6.71_{\pm0.04}$ | $6.63_{\pm0.03}$ |
| | QERA-approx | 21.68 | 20.52 | 23.31 | 21.83 | 11.15 | 10.99 | 9.11 | 9.04 | 12.51 | 11.72 | 6.77 | 6.64 |
| | **w/ SRR** | $20.53_{\pm0.01}$ | $19.39_{\pm0.04}$ | $22.68_{\pm0.13}$ | $19.16_{\pm0.15}$ | $10.91_{\pm0.01}$ | $10.69_{\pm0.00}$ | $9.05_{\pm0.01}$ | $8.92_{\pm0.01}$ | $12.06_{\pm0.00}$ | $11.43_{\pm0.01}$ | $6.70_{\pm0.01}$ | $6.60_{\pm0.01}$ |
| | QERA-exact | 20.10 | 19.59 | 20.10 | 19.36 | 10.84 | 10.68 | 9.04 | 8.97 | 11.37 | 11.00 | 6.68 | 6.55 |
| | **w/ SRR** | $19.62_{\pm0.02}$ | $18.71_{\pm0.04}$ | $19.33_{\pm0.04}$ | $18.30_{\pm0.05}$ | $10.76_{\pm0.00}$ | $10.59_{\pm0.00}$ | $9.00_{\pm0.00}$ | $8.91_{\pm0.01}$ | $11.24_{\pm0.00}$ | $10.78_{\pm0.01}$ | $6.63_{\pm0.01}$ | $6.46_{\pm0.00}$ |

*Table 1.* Perplexity ($\downarrow$) on WikiText2 under 3-bit MXINT quantization, evaluated with low rank settings $r=32$ and $r=64$. SRR is applied to three QER methods across six models. Lowest perplexity values are highlighted in **bold**. For SRR, results are reported as mean $\pm$ std over three random seeds. All evaluations are conducted using `lm-eval` framework.

| Method | TinyLlama 1.1B | Gemma-2 2B | LLaMA-2 7B | LLaMA-2 13B | LLaMA-3.1 8B | LLaMA-3.1 70B |
|---|---|---|---|---|---|---|
| BF16 | 46.38 | 59.26 | 58.90 | 63.32 | 67.34 | 74.83 |
| *w-only* | 42.01 | 45.12 | 52.50 | 56.98 | 51.17 | 66.02 |
| QERA-exact | 45.15 | 52.15 | 55.28 | 60.48 | 59.05 | 71.08 |
| **w/ SRR** | **46.79** | **54.38** | **56.56** | **61.58** | **60.79** | **71.39** |

*Table 2.* Average zero-shot accuracy ($\uparrow$) on five downstream tasks under low-rank setting $r=64$, using 3-bit MXINT quantizer. SRR is applied on QERA-exact, and best results are highlighted in **bold**. Full results and other setups are provided in Appendix C.1.

where $\gamma \in (0,1)$ controls the attenuation strength; updates to the residual components $\mathbf{L}^{(2)}, \mathbf{R}^{(2)}$ are unscaled.

In our experiments, coarse choices $\gamma \in \{0.1, 0.5\}$ all work well, indicating that QPEFT is not sensitive to the exact value of $\gamma$ as long as updates to $\mathbf{L}^{(1)}\mathbf{R}^{(1)}$ are attenuated relative to $\mathbf{L}^{(2)}\mathbf{R}^{(2)}$. This indicates that SRR's gains in QPEFT primarily arise from a better initialization, while gradient scaling further serves as a simple regularizer that prevents drift in the preserved subspace. Empirically, once two components are decoupled, a range of reasonable scaling strategies consistently outperform prior QPEFT baselines.

# 5. Experiments

In this section, we evaluate SRR on PTQ and QPEFT across models, bitwidths, and baselines. Experimental details are provided in Appendix A. Note that SRR introduces additional SVDs compared to the QER approach, but incurs only minimal computational overhead by using randomized SVD (Halko et al., 2011), as only a small number of leading singular values are required (e.g., a 1.06× increase for QERA-exact on LLaMA-2 7B). Details of computational and memory overhead are provided in Appendix A.4.

## 5.1. Experiments on PTQ

**Setup.** We apply SRR on top of existing QER methods to evaluate its effectiveness across different PTQ approaches. Each QER baseline defines a different scaling matrix $\mathbf{S}$, allowing us to examine SRR's consistency across diverse formulations. We compare against three baselines: two heuris-

tic scaling methods, LQER (Zhang et al., 2024a) and QERA-approx, and one exact solution, QERA-exact (Zhang et al., 2025). Another exact formulation, CALDERA (Saha et al., 2024), recovers the same solution in the unconstrained setting under an identical calibration set when no iterative refinement is applied; therefore, we report QERA-exact as a representative exact baseline. For reference, we include BF16 (full precision) and *w-only* quantization baselines. To ensure fair and stable evaluation, we repeat SRR with three different random seeds, as the rank split $k$ is selected using a randomized construction of the quantization error matrix, and report the mean and standard deviation across runs.

We evaluate a range of model sizes, including TinyLlama-1.1B (Zhang et al., 2024b), Gemma-2 2B (Team et al., 2024), LLaMA-2 7B and 13B (Touvron et al., 2023), and LLaMA-3.1 8B and 70B (Grattafiori et al., 2024). All models are primarily quantized to 3-bit precision using MX-INT (Darvish Rouhani et al., 2023) with a block size of 32, and we consider low-rank settings of $r=32$ and $r=64$.

We report perplexity on WikiText2 (Merity et al., 2017) and zero-shot accuracy on five downstream tasks (task details in Appendix A.2). All evaluations use the `lm-eval` framework following Zhang et al. (2025) to ensure consistency.

**Perplexity Reduction and Improved Zero-shot Accuracy.** As shown in Table 1, SRR consistently reduces perplexity across diverse models at $r=32$ and $r=64$, achieving relative reductions of up to 12.2% on Gemma-2 2B, 27.1% on LLaMA-2 7B, and 3.6% on LLaMA-3.1 8B.

For zero-shot accuracy on downstream tasks, SRR consistently outperforms the corresponding QER baselines across all model scales, as shown in Table 2. These improvements indicate that even with exact solutions such as QERA-exact for layer-wise reconstruction, there remains room for further enhancement through a structured low-rank term.

Overall, these results show that balancing structure preservation and error reconstruction enables more effective use of a fixed low-rank budget, improving reconstruction fidelity. Additional experimental results are provided in Appendix C.

| | | Method | Rank | MNLI Acc. | QNLI Acc. | RTE Acc. | SST Acc. | MRPC Acc. | CoLA Matt. | QQP Acc. | STSB P/S Corr. | Avg. |
|---|---|---|---|---|---|---|---|---|---|---|---|---|
| Quantization Bits | 16 | Full FT | – | 87.62±0.04 | 93.03±0.04 | 76.53±0.36 | 95.18±0.11 | 89.95±0.25 | 61.79±0.43 | 91.55±0.02 | 90.28±0.13 / 90.05±0.14 | 85.73 |
| | | LoRA | 8 | 87.59±0.04 | 92.68±0.05 | 72.80±0.21 | 95.07±0.11 | 89.79±0.14 | 61.08±0.54 | 90.95±0.03 | 90.09±0.16 / 89.84±0.15 | 84.99 |
| | 4.25 | QLoRA | | 86.91±0.08 | 92.29±0.10 | 66.06±0.63 | 94.15±0.20 | 86.76±0.49 | 56.24±1.32 | 90.45±0.08 | 88.95±0.39 / 88.82±0.41 | 82.72 |
| | | LoftQ | | 87.13±0.08 | 91.63±0.09 | 64.26±0.36 | 93.46±0.11 | 87.75±0.42 | 59.07±1.20 | 90.46±0.07 | 88.95±0.35 / 88.84±0.34 | 82.83 |
| | | QERA | 8 | 87.07±0.07 | 92.20±0.09 | 64.98±0.36 | 94.15±0.11 | 87.99±0.39 | 58.55±1.14 | 90.45±0.07 | 89.86±0.33 / 89.68±0.31 | 83.15 |
| | | LQ-LoRA | | 85.89±0.09 | 90.96±0.11 | 54.15±0.63 | 92.32±0.20 | 82.35±0.49 | 42.60±1.44 | 88.67±0.09 | 85.89±0.42 / 85.73±0.44 | 77.84 |
| | | **SRR** | | **87.15**±0.07 | **92.67**±0.08 | **72.68**±0.42 | **94.27**±0.11 | **89.71**±0.46 | **60.07**±1.02 | **90.49**±0.06 | **90.00**±0.30 / **89.82**±0.28 | **84.62** |
| | 3.25 | QLoRA | | 86.14±0.13 | 90.76±0.17 | 54.87±0.72 | 90.83±0.23 | 78.92±0.72 | 10.83±2.11 | 89.91±0.13 | 86.77±0.59 / 86.28±0.62 | 73.60 |
| | | LoftQ | | 86.38±0.12 | 90.24±0.15 | 57.04±0.72 | 91.63±0.23 | 81.13±0.74 | 14.52±1.92 | 89.27±0.12 | 86.55±0.56 / 86.24±0.54 | 74.58 |
| | | QERA | 8 | 86.49±0.11 | 89.46±0.15 | 57.40±0.72 | 91.74±0.23 | 84.56±0.65 | 28.98±1.82 | 89.26±0.11 | 87.90±0.53 / 87.61±0.55 | 76.96 |
| | | LQ-LoRA | | 84.70±0.14 | 88.74±0.18 | 54.51±0.96 | 91.63±0.30 | 74.75±0.65 | 24.37±2.30 | 87.61±0.14 | 85.16±0.67 / 85.31±0.64 | 73.94 |
| | | **SRR** | | **86.54**±0.10 | **91.48**±0.13 | **62.82**±0.63 | **93.12**±0.23 | **87.75**±0.65 | **53.17**±1.63 | **90.05**±0.11 | **88.18**±0.48 / **87.82**±0.46 | **81.62** |
| | 2.25 | QLoRA | | 75.29±0.25 | 83.55±0.33 | 52.71±2.37 | 87.50±0.50 | 68.87±2.34 | 0.00±0.00 | 87.73±0.25 | 58.23±1.85 / 57.90±1.92 | 64.21 |
| | | LoftQ | | 80.52±0.23 | 83.93±0.29 | 50.18±1.30 | 89.68±0.46 | 68.95±1.35 | 0.00±0.00 | 87.95±0.23 | 82.51±1.05 / 80.17±1.05 | 67.82 |
| | | QERA | 64 | 82.41±0.22 | 86.08±0.28 | 54.39±1.27 | 90.94±0.41 | 74.75±1.30 | 18.72±3.42 | 89.46±0.22 | 84.12±1.00 / 82.50±0.93 | 72.51 |
| | | LQ-LoRA | | 82.07±0.28 | 85.22±0.35 | 51.99±1.65 | 88.72±0.54 | 68.38±1.61 | 0.00±0.00 | 88.20±0.28 | 77.27±1.26 / 77.80±1.19 | 67.76 |
| | | **SRR** | | **84.63**±0.19 | **89.82**±0.28 | **58.84**±1.08 | **92.09**±0.40 | **86.27**±1.12 | **39.48**±5.00 | **90.02**±0.19 | **86.46**±0.84 / **86.06**±0.89 | **78.43** |

*Table 3.* Fine-tuning results on the GLUE using RoBERTa-base under 4/3/2-bit MXINT quantizer. All methods are trained for 5 epochs using best-performing learning rates (see Appendix A.3). Results are reported as mean ± std over runs with best results in **bold**.

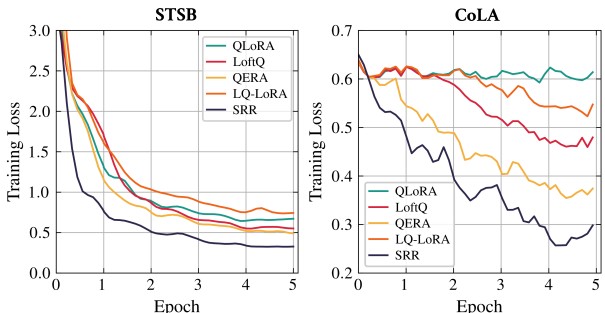

*Figure 4.* Training loss curves for QPEFT baselines on two GLUE tasks: STSB (**Left**) and CoLA (**Right**), over five epochs. Each method uses its best-performing learning rate (see Appendix A.3). SRR shows faster training loss reduction than the other baselines.

## 5.2. Experiments on QPEFT

**Setup.** We use the SRR-initialized low-rank adapter with fixed gradient scaling $\gamma = 0.1$ (Equation (7)) as the default setting and compare against four baselines: QLoRA (Dettmers et al., 2023), LoftQ (Li et al., 2024), QERA (Zhang et al., 2025), and LQ-LoRA (Guo et al., 2024). For consistency, all methods that rely on activation statistics (QERA, LQ-LoRA, and SRR) share the same scaling **S** from QERA-exact, which yields an exact layer-wise reconstruction solution. For iterative methods (LoftQ and LQ-LoRA), the results are reported after five iterations.

We evaluate RoBERTa-base (Liu et al., 2019) on GLUE (Wang et al., 2019) and LLaMA-2 7B / LLaMA-3.1 8B on both SlimPajama (Soboleva et al., 2023) and GSM8K (Cobbe et al., 2021), following Zhang et al. (2025). The MXINT quantizer (Darvish Rouhani et al., 2023) is used throughout, with 4/3/2-bit precision for GLUE, respectively, and 4/2-bit precision for SlimPajama and GSM8K. Additional experimental details, including the truncated SVD factorization scheme, are provided in Appendix A.3.

**Improved QPEFT.** As shown in Table 3, SRR delivers consistent accuracy improvements across all quantization levels. Without any iterative refinement, SRR outperforms iterative methods like LoftQ and LQ-LoRA by over 10 percentage points at 2-bit precision. Compared to the recent QERA baseline, SRR achieves gains of 1.5, 4.7, and 5.9 percentage points at 4-bit, 3-bit, and 2-bit settings, respectively. These improvements are most significant in low-bit regimes, where aggressive quantization typically causes severe performance degradation. Beyond accuracy, SRR accelerates convergence during training, with loss decreasing faster than baseline methods (Figure 4). See Appendix F for training-loss curves on all GLUE tasks, including 20 epoch curves.

| | | Method | LLaMA-2 7B | | LLaMA-3.1 8B | |
|---|---|---|---|---|---|---|
| | | | SlimPajama PPL (↓) | GSM8K Acc (↑) | SlimPajama PPL (↓) | GSM8K Acc (↑) |
| Quantization Bits | 16 | LoRA | 6.27±0.05 | 35.41±0.30 | 8.11±0.06 | 57.24±0.21 |
| | 4.25 | QLoRA | 6.48±0.03 | 32.21±0.25 | 8.78±0.04 | 54.20±0.22 |
| | | LoftQ | 6.52±0.06 | 28.35±0.81 | 8.88±0.08 | 54.16±0.49 |
| | | QERA | 6.49±0.06 | 32.13±0.51 | 8.77±0.09 | 54.06±0.28 |
| | | LQ-LoRA | 6.66±0.08 | 29.82±0.42 | 9.59±0.11 | 48.16±0.56 |
| | | **SRR** | **6.46**±0.05 | **34.18**±0.37 | **8.68**±0.05 | **54.34**±0.17 |
| | 2.25 | QLoRA | 11.84±0.20 | 13.59±1.06 | 24.46±0.37 | 20.02±0.84 |
| | | LoftQ | 11.74±0.14 | 14.67±0.88 | 23.26±0.19 | 21.53±0.73 |
| | | QERA | 10.87±0.09 | 16.83±0.43 | 21.94±0.14 | 25.11±0.59 |
| | | LQ-LoRA | 11.46±0.13 | 15.91±0.57 | 22.07±0.22 | 24.16±0.45 |
| | | **SRR** | **10.72**±0.12 | **18.29**±0.69 | **19.62**±0.18 | **26.87**±0.61 |

*Table 4.* Fine-tuning results on SlimPajama with perplexity ($r$=8) and GSM8K with accuracy ($r$=64) under 4/2-bit MXINT quantizer, using best learning rates (see Appendix A.3). SlimPajama is trained for 1000 steps and GSM8K for 10 epochs. Results are reported as mean ± std over three runs with best results in **bold**.

SRR also excels on SlimPajama and GSM8K, outperforming all baselines in both perplexity and accuracy (Table 4). At 2-bit precision, SRR reduces SlimPajama perplexity while improving GSM8K accuracy by over 1.4 percentage points on LLaMA-2 7B and over 1.7 percentage points

on LLaMA-3.1 8B compared to QERA. These consistent gains demonstrate that SRR's structured decomposition provides a more effective initialization for low-rank adapters, by preserving dominant directions and stabilizing them via gradient scaling, thereby enabling effective application of SRR to QPEFT even under aggressive quantization.

# 6. Analysis

**Distribution of the Selected Rank $k^\star$.** We analyze how the selected rank $k^\star$ is distributed across projection types in LLaMA-2 7B and LLaMA-3.1 8B, using QERA-exact for rank selection. We find that the selected ranks vary systematically by weight matrix and their associated scaling, reflecting differences in the underlying structure (see Figure 5). These results support the need for SRR, which adaptively selects the preserved rank $k^\star$ for each weight-scale pair. Additional analyses for $k^\star$ are provided in Appendix B.2.

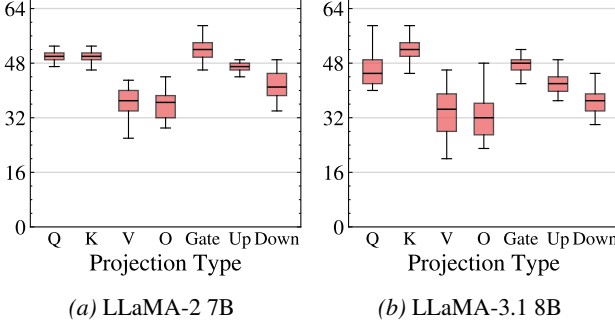

*(a)* LLaMA-2 7B          *(b)* LLaMA-3.1 8B

*Figure 5.* Projection-wise distribution of the selected rank $k^\star$ under a rank budget of $r = 64$. Each box plot summarizes layer-wise variations for (a) LLaMA-2 7B and (b) LLaMA-3.1 8B.

**Applying SRR on Other Quantizers.** To assess whether SRR is quantizer-agnostic, we apply it to the 3-bit GPTQ (Frantar et al., 2023) quantizer as well as the 2-bit QuIP# (Tseng et al., 2024) quantizer, and evaluate perplexity on LLaMA-2 7B and LLaMA-3.1 8B. As shown in Table 5, SRR consistently reduces perplexity for all QER-based methods under both quantization schemes. These results demonstrate that SRR provides reliable improvements even under aggressive low-bit quantization, indicating broad compatibility with different quantization schemes. Note that, with the same calibration set, SRR selects the preserved rank $k^\star$ in a quantizer-agnostic manner, yielding identical rank allocations across quantizers. Implementation details and quantizer-specific settings are provided in Appendix A.2.

**Gradient Scaling on the Preserved Top-$k^\star$ Directions.** To examine how gradient scaling on the preserved top-$k^\star$ directions affects QPEFT, we compare fixed scaling factors $\gamma \in \{0, 0.1, 0.5, 1\}$ with SGP (Saha & Roy, 2023), a gradient-scaling strategy originally proposed for continual

| Method | Rank | LLaMA-2 7B | | LLaMA-3.1 8B | |
|---|---|---|---|---|---|
| | | GPTQ (3-bit) | QuIP# (2-bit) | GPTQ (3-bit) | QuIP# (2-bit) |
| BF16 | - | 8.71 | | 7.55 | |
| *w-only* | | 10.29 | 15.97 | 9.69 | 16.59 |
| LQER | 64 | 10.07 | 15.53 | 9.48 | 15.28 |
| **w/ SRR** | | **10.05**±0.00 | **14.91**±0.04 | **9.47**±0.00 | **14.73**±0.03 |
| QERA-approx | | 10.07 | 15.20 | 9.50 | 15.16 |
| **w/ SRR** | | **10.01**±0.01 | **14.62**±0.03 | **9.44**±0.00 | **14.67**±0.01 |
| QERA-exact | | 10.06 | 15.03 | 9.42 | 14.84 |
| **w/ SRR** | | **9.98**±0.01 | **14.55**±0.02 | **9.37**±0.01 | **13.73**±0.02 |

*Table 5.* Perplexity (↓) on WikiText2, using 3-bit GPTQ and 2-bit QuIP# quantizers. SRR is applied on QER methods. All evaluations are performed using `lm-eval`, with best results in **bold**.

learning. SGP selectively attenuates gradient components along high-importance directions; under SRR, these directions naturally correspond to the preserved dominant subspace spanned by the top-$k^\star$ singular vectors. In all cases, we scale only the gradients on the preserved directions and leave the residual directions unchanged. As shown in Table 6, both extreme choices degrade performance: without attenuation ($\gamma = 1$), updates drift into the preserved subspace and weaken the SRR initialization, whereas full suppression ($\gamma = 0$) over-constrains the preserved component and limits adaptation. In contrast, moderate scaling ($\gamma = 0.1$ or $0.5$), as well as SGP, consistently achieves strong and comparable performance. Overall, these results indicate that coarse attenuation of the preserved directions is sufficient, and that the gains in SRR-based QPEFT primarily arise from the SRR-initialized decomposition rather than fine-grained scaling. Refer to Appendix A.3 for SGP details and to Appendix D for per-task results, and additional SGP comparisons.

| Bits | Rank | Gradient Scaling | | | | |
|---|---|---|---|---|---|---|
| | | $\gamma = 0$ | $\gamma = 1$ | $\gamma = 0.5$ | $\gamma = 0.1$ | SGP ($\alpha = 5$) |
| 4.25 | 8 | 80.18 | 81.70 | 83.78 | **84.62** | 84.42 |
| 2.25 | 64 | 71.10 | 73.38 | 77.58 | 78.43 | **78.88** |

*Table 6.* Average zero-shot accuracy (↑) of SRR-based QPEFT on GLUE for RoBERTa-base, comparing top-$k^\star$ gradient scaling settings ($\gamma \in \{0, 0.1, 0.5, 1\}$ and SGP ($\alpha = 5$)). Best results are highlighted in **bold**, with the second-best results underlined. Full results are provided in Appendix D.

# 7. Conclusion

We proposed **Structured Residual Reconstruction (SRR)**, a *preserve-then-quantize* framework that explicitly allocates rank to balance subspace preservation and quantization error correction under a fixed rank budget. Guided by a theory-based criterion for selecting the balancing rank $k^\star$, SRR yields a more accurate approximation of the form $\mathbf{W} \approx \mathbf{Q} + \mathbf{LR}$ than prior QER methods. For QPEFT, gradient scaling on preserved directions stabilizes fine-tuning and focuses learning on residual components. Experiments on PTQ and QPEFT consistently demonstrate the effectiveness of SRR across models and settings, establishing it as a principled framework for low-rank refinement in quantized models.

## Impact Statement

This work improves the efficiency of quantized models through principled rank allocation, enabling more accurate low-rank reconstruction under fixed resource budgets. While it can reduce the cost of deploying large models, it introduces no new ethical risks beyond those already associated with prior quantization and model compression methods.

## Acknowledgement

This work was supported by Institute of Information & communications Technology Planning & Evaluation (IITP) grant funded by the Korea government(MSIT) (No. RS-2024-00457882, National AI Research Lab Project), the Ministry of Science and ICT (MSIT), South Korea, under the Information Technology Research Center (ITRC) Support Program (IITP-2025-RS-2022-00156295), and K-CHIPS (Korea Collaborative & High-tech Initiative for Prospective Semiconductor Research) (RS-2024-00405946, 24052-15TC) funded by the Ministry of Trade, Industry & Energy (MOTIE, Korea).

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

# A. Experiment Details

## A.1. Experimental Setup

Here, we list the models used in our experiments, along with their sources and licensing information, respectively in Table 7. Besides, PTQ and QPEFT experiments were executed using both 8 NVIDIA A100 and 8 NVIDIA L40S GPUs, distributed across separate machines. Our experiments required approximately 9,000 GPU hours for PTQ and 5,000 GPU hours for QPEFT, totaling 14,000 GPU hours.

| Model | Source | License |
|---|---|---|
| RoBERTa-base | (Liu et al., 2019) | MIT License |
| TinyLlama-1.1B | (Zhang et al., 2024b) | Apache License 2.0 |
| Gemma-2-2B | (Team et al., 2024) | Gemma License |
| LLaMA-2-7B | (Touvron et al., 2023) | LLaMA 2 Community License |
| LLaMA-2-13B | (Touvron et al., 2023) | LLaMA 2 Community License |
| LLaMA-3.1-8B | (Grattafiori et al., 2024) | LLaMA 3.1 Community License |
| LLaMA-3.1-70B | (Grattafiori et al., 2024) | LLaMA 3.1 Community License |

*Table 7.* Summary of models used in this paper, including source, and license.

## A.2. Experiment Details for PTQ

**Setup and Quantization Configuration.** We evaluate our method in the post-training quantization (PTQ) setting across various scales, including TinyLlama-1.1B (Zhang et al., 2024a), Gemma-2 2B (Team et al., 2024), LLaMA-2 7B, 13B (Touvron et al., 2023), and LLaMA-3.1 8B, 70B (Grattafiori et al., 2024). Weights are quantized to 3-bit precision using MXINT (Darvish Rouhani et al., 2023) with block size 32, yielding effective bitwidth of 3.25. Low-rank correction terms are computed with rank 32 and 64 for 3-bit quantization.

**Evaluation Benchmarks.** To assess performance comprehensively, we report both language modeling perplexity and downstream task accuracy. Perplexity is evaluated on WikiText2 (Merity et al., 2017) using `lm-eval` (Gao et al., 2024). For downstream evaluation, we conduct zero-shot evaluation on five benchmarks designed to test reasoning and factual understanding: HellaSwag (Zellers et al., 2019), Winogrande (Sakaguchi et al., 2020), BoolQ (Clark et al., 2019), MMLU (Hendrycks et al., 2021), and BigBench-Hard (Suzgun et al., 2023). Task descriptions are summarized in Table 8.

| Task | Description |
|---|---|
| HellaSwag | Choose the most plausible continuation for a given scenario |
| Winogrande | Resolve ambiguous pronouns using commonsense reasoning |
| BoolQ | Is the answer to the question supported by the paragraph? |
| MMLU | Answer multiple-choice questions across 57 academic subjects |
| BBH | Assesses advanced reasoning capabilities on challenging problems |

*Table 8.* Descriptions of five downstream tasks used in our PTQ evaluation.

**Baselines.** We evaluate our approach against leading PTQ baselines that incorporate low-rank quantization error reconstruction, including LQER (Zhang et al., 2024a) and QERA (Zhang et al., 2025) in both its approximate and exact forms. All baseline implementations are standardized to use the same quantization format, block size, and calibration data to ensure fair comparison. In addition, we include quantization-only models (*w-only*) to isolate the effect of low-rank correction. For LQER, QERA-approx, and QERA-exact, we use a calibration set of 256 samples from SlimPajama-6B (Soboleva et al., 2023).

**Additional Quantizer Settings** For completeness, we summarize the quantizer-specific configurations used in the experiments reported in Table 5. For GPTQ (Frantar et al., 2023), we adopt a 3-bit quantization setting with a group size of 128. The quantization is performed using asymmetric quantization, a damping factor of 0.01 with an automatic increment of 0.0025, and blockwise column quantization. The Hessian matrix required for GPTQ is estimated using 256 samples from

the C4 dataset (Dodge et al., 2021). For QuIP# (Tseng et al., 2024), we follow the original quantization setup and employ an offline Hessian estimated using 256 samples from the RedPajama dataset (Weber et al., 2024).

### A.3. Experiment Details for QPEFT

**Configuration Before Fine-tuning.** Unless stated otherwise, we use the SRR-initialized low-rank adapter with fixed gradient scaling $\gamma = 0.1$ (Equation (7)) as a default setting. Baselines include QLoRA (Dettmers et al., 2023), LoftQ (Li et al., 2024), QERA (Zhang et al., 2025), and LQ-LoRA (Guo et al., 2024). In GLUE tasks, we quantize weights with MXINT using block size 32: 4-bit and 3-bit settings use **LR** rank 8, while the 2-bit setting uses **LR** rank 64. Here, MXINT's block-wise shared exponent yields average (effective) bitwidths of 4.25, 3.25, and 2.25 bits for the 4/3/2-bit configurations, respectively. In GSM8K tasks, weights are quantized using the same 2-bit and 4-bit configurations, both with **LR** rank 64. For SlimPajama, we apply the same quantization settings with **LR** rank 8. Additionally, we adopt five iterations for LoftQ (Li et al., 2024) and LQ-LoRA (Guo et al., 2024). For QERA (Zhang et al., 2025), we consistently adopt the exact scaling mode (QERA-exact). While LQ-LoRA originally applies a Fisher-weighted objective to determine scaling directions, we instead adopt the exact same scaling scheme as QERA-exact for a fair comparison. In addition, we do not use the dynamic quantization configurations employed by LQ-LoRA. These choices provide a unified activation-aware basis across baselines and eliminate confounding factors arising from inconsistent scaling heuristics. Calibration is performed using WikiText-2-raw-v1 (Merity et al., 2017) for GLUE, and SlimPajama-100M (iankur, 2024) for SlimPajama and GSM8K.

**Fine-tuning on Natural Understanding Tasks: GLUE.** We evaluate our approach on the GLUE benchmark (Wang et al., 2019), which comprises eight diverse tasks: MNLI (Williams et al., 2018), QNLI (Wang et al., 2019), RTE (Dagan et al., 2006), SST-2 (Socher et al., 2013), MRPC (Dolan & Brockett, 2005), QQP (Wang et al., 2019), CoLA (Warstadt et al., 2019), and STSB (Cer et al., 2017), see task descriptions in Table 9. Following prior work (Zhang et al., 2025), we report accuracy for MNLI, QNLI, RTE, SST-2, MRPC, and QQP; Matthews correlation for CoLA; and Pearson/Spearman correlations for STSB. All methods are built upon RoBERTa-base (Liu et al., 2019) and fine-tuned using a consistent strategy, where each model is trained for 5 epochs. To ensure fair comparisons, task-specific learning rates are selected, with details provided in Table 10. All results are reported as mean $\pm$ std over three runs with different random seeds to ensure robustness.

| Task | Description |
|------|-------------|
| MNLI | Infer relation: entailment / neutral / contradiction |
| QNLI | Does the context sentence answer the question? |
| RTE | Does the premise entail the hypothesis? |
| SST-2 | Sentiment classification (positive/negative) |
| MRPC | Are the two sentences paraphrases? |
| CoLA | Grammatical acceptability |
| QQP | Are the two questions semantically equivalent? |
| STSB | Predict semantic similarity score (0–5) between sentences |

*Table 9.* Descriptions of the eight GLUE tasks used in our evaluation.

| Bits | Rank | Method | Learning Rates |
|------|------|--------|----------------|
| 16 | – | Full FT | 7e-5, 5e-5, 3e-5, 2e-5 |
| 16 | 8 | LoRA | 3e-4, 5e-4, 6e-4, 7e-4 |
| 4.25 | 8 | QLoRA / LoftQ / QERA / LQ-LoRA / SRR | 5e-5, 7e-5, 1e-4, 3e-4, 5e-4, 6e-4, 7e-4 |
| 3.25 | 8 | QLoRA / LoftQ / QERA / LQ-LoRA / SRR | 3e-5, 5e-5, 7e-5, 1e-4, 3e-4, 5e-4, 6e-4 |
| 2.25 | 64 | QLoRA / LoftQ / QERA / LQ-LoRA / SRR | 1e-5, 3e-5, 5e-5, 7e-5, 9e-5, 1e-4, 2e-4 |

*Table 10.* Learning rates of RoBERTa-base experiments on GLUE.

**Fine-tuning on Language Modeling and Reasoning Tasks: SlimPajama and GSM8K.** To assess generative capabilities, we fine-tune LLaMA-2 7B (Touvron et al., 2023) and LLaMA-3.1 8B (Grattafiori et al., 2024) on SlimPajama (Soboleva et al., 2023), a corpus for general language modeling framed as causal language modeling (CLM), and GSM8K (Cobbe et al., 2021), which tests arithmetic reasoning. We evaluate performance using perplexity on SlimPajama and exact-match

accuracy on GSM8K. For SlimPajama, training is conducted for 1,000 steps with a total batch size of 16. We sweep learning rates over $9e-5, 1e-4, 3e-4, 5e-4$ for standard methods and $5e-5, 7e-5, 9e-5, 1e-4$ for LQ-LoRA. The best-performing configuration for each method is reported. For GSM8K, models are trained for 10 epochs with a total batch size of 32. Learning rates are swept over $7e-5, 9e-5, 1e-4, 3e-4, 5e-4$ for all methods except LQ-LoRA, which uses $3e-5, 5e-5, 7e-5, 1e-4$. All reported results are averaged over three runs with different random seeds to ensure robustness.

**Implementation Details of SVD Decomposition.** Let $\mathbf{U}_r \boldsymbol{\Sigma}_r \mathbf{V}_r^\top$ denote a truncated singular value decomposition. This decomposition can be expressed in multiple equivalent low-rank forms $\mathbf{LR}$. Unless otherwise specified, we adopt the factorization $\mathbf{L} = \mathbf{U}_r$ and $\mathbf{R} = \boldsymbol{\Sigma}_r \mathbf{V}_r^\top$ in our experiments, which exactly represents the truncated SVD while keeping the left factor orthonormal.

**Rank-wise Gradient Scaling via SGP.** Beyond the fixed scaling rule in Equation (7), we also consider Singular Gradient Projection (SGP) (Saha & Roy, 2023), which is motivated by continual learning and selectively constrains updates along important subspaces. Equation (7) can be expressed in a rank-wise form by scaling gradient components along the preserved directions:

$$\mathbf{u}_i^\top \nabla_{\mathbf{L}^{(1)} \mathbf{R}^{(1)}} \mathcal{L} \; \leftarrow \; (1 - \lambda_i) \, \mathbf{u}_i^\top \nabla_{\mathbf{L}^{(1)} \mathbf{R}^{(1)}} \mathcal{L}, \qquad i \le k^\star, \tag{8}$$

where $\mathbf{u}_i$ denotes the $i$-th left singular vector of the preserved adapter $\mathbf{L}^{(1)} \mathbf{R}^{(1)}$, with corresponding singular value $\sigma_i$. SGP specifies an importance-aware choice of $\lambda_i \in [0, 1]$ as

$$\lambda_i = \frac{(\alpha + 1)\sigma_i}{\alpha \sigma_i + \sigma_1}, \tag{9}$$

with $\sigma_1$ being the largest singular value and $\alpha \ge 0$ controlling the overall scaling strength. Unless otherwise specified, we use the default setting $\alpha = 5$. Refer to Saha & Roy (2023) for further methodological details.

### A.4. Computational and Memory Overhead

| Method | Scaling | QER | | SRR | | SRR Overhead | |
|---|---|---|---|---|---|---|---|
| | *Time* | *Time* | *Total* | *Time* | *Total* | *QER vs. SRR* | *Full Pipeline* |
| QERA-exact | 808.85 | 10.20 | 819.05 | 10.82 | 819.67 | ×1.06 | ×1.00 |

*Table 11.* Computation time (in GPU minutes) on LLaMA-2 7B, measured using a single NVIDIA L40S GPU. **Scaling** denotes the time to compute the scaling matrix. **QER** and **SRR** report the time for quantization and reconstruction without and with SRR, respectively, with *Total* including scaling time. *QER vs. SRR* reports the overhead measured on the quantization and reconstruction stage, while *Full Pipeline* includes scaling and reflects the end-to-end overhead.

We analyze the computational overhead introduced by SRR when applied to QER-based methods. A standard QER pipeline consists of two main stages: (i) computing a scaling matrix from activation statistics and (ii) reconstructing quantized weights using a low-rank correction.

SRR augments this pipeline by introducing two additional singular value decompositions to identify the dominant subspaces of the scaled weight matrix $\mathbf{SW}$ and the scaled quantization error $\mathbf{SE}$. Importantly, SRR only requires the top-$r$ singular components of both $\mathbf{SW}$ and $\mathbf{SE}$, where $r$ denotes the total rank budget. Accordingly, both decompositions are implemented using randomized SVD (Halko et al., 2011), with the number of power iterations set to $n_{\mathrm{iter}} = 4$ and the oversampling parameter set to twice the target rank.

While a full SVD on an $m \times n$ matrix incurs a computational cost of $\mathcal{O}(mn^2)$, the randomized SVD used in SRR reduces the complexity to $\mathcal{O}(mnr)$, where $r$ denotes the target rank and satisfies $r \ll \min(m, n)$ in all our experiments. As a result, the additional SVD computations introduce negligible overhead relative to the overall pipeline. We further empirically verify that replacing full SVD with randomized SVD does not affect performance: across all evaluated settings, the resulting differences in perplexity compared to full SVD are negligible.

In practice, the dominant computational cost arises from computing the scaling matrix. Across our experiments, SRR increases the cost of the quantization and reconstruction stage by approximately $6\%$, while the end-to-end computation time remains effectively unchanged (within $1\%$). As shown in Table 11, this confirms that the additional overhead introduced by SRR is negligible compared to the cost of scaling.

Finally, SRR incurs no additional memory overhead in either PTQ or QPEFT settings, as it does not introduce extra low-rank parameters beyond those already used by the underlying QER method.

## B. Analysis of Rank Selection in SRR

### B.1. Stability of Rank Selection across Random Seeds

We first examine the stability of the proposed rank selection criterion with respect to randomness in the quantization error proxy. All experiments in this subsection are conducted on LLaMA-2 7B and LLaMA-3.1 8B using a fixed scaling matrix $\mathbf{S}$ and a fixed total rank budget of $r=64$. The only source of randomness is the random seed used to generate the proxy quantization error matrix $\mathbf{E}$.

For each layer, we evaluate the rank selection under two different random seeds and compute the corresponding optimal rank

$$k^\star = \arg\min_{0 \leq k \leq r} \rho_k(\mathbf{SW})\,\rho_{r-k}(\mathbf{SE}), \tag{10}$$

where $\rho_k(\cdot)$ denotes the spectral tail energy beyond rank $k$. Table 12 summarizes the resulting values of $k^\star$ across the two seeds for representative layers, reporting their mean and variation.

Across layers, the selected ranks are highly stable with respect to the choice of random seed. In most cases, the variation in $k^\star$ is limited to at most $\pm 1$ rank, and even in the worst case the deviation is bounded by three. The distribution of selected ranks is sharply concentrated around a single value, indicating that the rank allocation is primarily governed by the spectral structure of the scaled weight matrix $\mathbf{SW}$, rather than by incidental fluctuations in the proxy noise realization.

Importantly, we find that such small perturbations in the selected rank have a negligible effect on reconstruction quality and downstream perplexity. This explains the consistently small standard deviation of perplexity reported in Table 1, and demonstrates that the proposed rank allocation is not only stable in selection but also robust in terms of end-task performance.

| Model | Mean $|\Delta k^\star|$ | | | | | | | Max $|\Delta k^\star|$ | | | | | | |
|---|---|---|---|---|---|---|---|---|---|---|---|---|---|---|
| | Query | Key | Value | Output | Gate | Up | Down | Query | Key | Value | Output | Gate | Up | Down |
| LLaMA-2 7B | 0.1 | 0.1 | 0.8 | 0.7 | 0.4 | 0.6 | 0.9 | 1 | 1 | 2 | 2 | 2 | 2 | 3 |
| LLaMA-3.1 8B | 0.4 | 0.3 | 0.2 | 0.5 | 0.2 | 0.2 | 0.3 | 2 | 1 | 1 | 2 | 1 | 1 | 1 |

*Table 12.* Stability of the selected rank $k^\star$ under two different random seeds. For each model and module type, we compute $|k^\star_{\text{seed1}} - k^\star_{\text{seed2}}|$ and report the mean and max absolute differences across two seeds. All results are obtained using QERA-exact with a 3-bit MXINT quantizer and a fixed total rank budget of $r=64$.

### B.2. Projection-wise Analysis of Rank Allocation

Figure 5 shows that the selected rank $k^\star$ varies substantially across projection types under a fixed rank budget of $r = 64$. We consider the query (Q), key (K), value (V), output (O), gate, up, and down projections in the attention and MLP blocks. Even when projections share identical inputs and activation-aware scaling, SRR assigns different preserved ranks, indicating that rank allocation is driven primarily by the structural properties of the weight matrices. This behavior is most clearly illustrated by the Q, K, and V projections.

The Q, K, and V projections use the same scaling matrix, yet their selected ranks differ markedly. The Q and K matrices directly participate in attention score computation via inner products, and therefore tend to exhibit more concentrated spectral structure with dominant singular directions (Yuan et al., 2023b; Wang et al., 2025). SRR accordingly allocates a larger preserved rank to Q and K to protect these low-rank components from quantization. In contrast, the V projection mainly supports content aggregation after attention weighting. Its representations are more distributed, leading to a flatter spectrum and a smaller preserved rank. As a result, SRR allocates a larger portion of the rank budget to reconstructing quantization-induced error for V. These projection-specific characteristics align closely with the selected ranks $k^\star$, indicating that SRR effectively captures underlying structural differences when allocating rank.

Consistent variation of $k^\star$ across the remaining projections further highlights the importance of structure-aware rank allocation, demonstrating that SRR adapts rank splits based on intrinsic weight structure rather than shared scaling statistics.

### B.3. Additional results on the alignment between reconstruction error and the rank-selection objective

Figure 6 presents results across different matrices, comparing the true reconstruction error with our rank-selection objective. The consistent trends of both quantities across $k$ support the use of the surrogate objective for selecting $k^\star$ in practice.

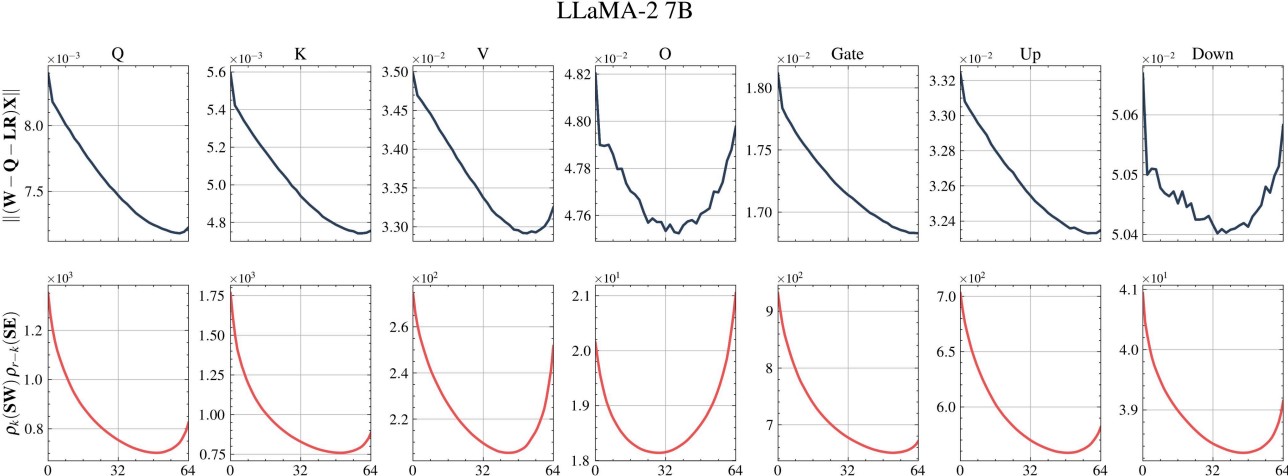

*Figure 6.* Alignment between reconstruction error and the rank-selection objective in LLaMA-2 7B (layer 10). Reconstruction error (**Top**) and surrogate objective (**Bottom**) as functions of the preserved rank $k$ under a rank budget of $r = 64$.

## C. Additional PTQ Experiments and Analysis

### C.1. Full Zero-shot Accuracy under 3-bit Quantization.

Table 13 reports the average zero-shot accuracy across five benchmarks under 3-bit PTQ, covering all models and the two adapter ranks ($r$=32 and $r$=64), while Table 14 presents the full per-task results. Overall, the results demonstrate that SRR consistently improves performance over QERA-exact and *w-only* baselines.

| | Method | TinyLlama 1.1B | Gemma-2 2B | LLaMA-2 7B | LLaMA-2 13B | LLaMA-3.1 8B | LLaMA-3.1 70B |
|---|---|---|---|---|---|---|---|
| | BF16 | 46.38 | 59.26 | 58.90 | 63.32 | 67.34 | 74.83 |
| | *w-only* | 42.01 | 45.12 | 52.50 | 56.98 | 51.17 | 66.02 |
| $r = 32$ | QERA-exact | 45.54 | 51.28 | 55.05 | 60.46 | 58.17 | 70.54 |
| | **w/ SRR** | **46.05** | **52.19** | **55.65** | **61.55** | **59.82** | **70.84** |
| $r = 64$ | QERA-exact | 45.15 | 52.15 | 55.28 | 60.48 | 59.05 | 71.08 |
| | **w/ SRR** | **46.79** | **54.38** | **56.56** | **61.58** | **60.79** | **71.39** |

*Table 13.* Average zero-shot accuracy ($\uparrow$) on five downstream tasks under two low-rank settings $r$=32 and $r$=64, using 3-bit MXINT quantizer. SRR is applied on QERA-exact, and best results are highlighted in **bold**.

### C.2. Performance on ZeroQuant-V2.

When SRR is applied to ZeroQuant-V2 (Yao et al., 2022), it assumes an identity scaling matrix, i.e., $\mathbf{S} = \mathbf{I}$. As shown in Figure 7, SRR consistently achieves lower weight reconstruction error, measured by $\|\mathbf{W} - \mathbf{Q} - \mathbf{LR}\|_F$, across all layers compared to existing QER frameworks. This result confirms the effectiveness of SRR in more accurately approximating the original weight matrix under the same rank budget.

| | Method | HellaSwag Acc_norm | | Winogrande Acc | | BoolQ Acc | | MMLU Acc | | BBH Acc_norm | |
|---|---|---|---|---|---|---|---|---|---|---|---|
| | | $r=32$ | $r=64$ | $r=32$ | $r=64$ | $r=32$ | $r=64$ | $r=32$ | $r=64$ | $r=32$ | $r=64$ |
| **TinyLlama 1.1B** | BF16 | 61.53 | | 59.27 | | 55.93 | | 25.25 | | 29.92 | |
| | *w-only* | 43.41 | | 50.91 | | 60.80 | | 25.70 | | 29.23 | |
| | QERA-exact | 55.71 | 56.37 | 57.38 | 55.96 | 60.37 | 59.05 | 24.39 | 24.58 | 29.86 | 29.79 |
| | w/ SRR | **55.88**$_{\pm0.33}$ | **57.18**$_{\pm0.36}$ | **58.01**$_{\pm0.41}$ | **59.33**$_{\pm0.64}$ | **60.53**$_{\pm0.73}$ | **61.48**$_{\pm0.84}$ | **25.59**$_{\pm0.29}$ | **25.76**$_{\pm0.67}$ | **30.22**$_{\pm0.10}$ | **30.19**$_{\pm0.64}$ |
| **Gemma-2 2B** | BF16 | 72.92 | | 68.59 | | 72.48 | | 49.23 | | 33.10 | |
| | *w-only* | 53.53 | | 60.62 | | 52.14 | | 28.35 | | 30.97 | |
| | QERA-exact | 64.31 | 65.21 | 63.30 | 62.75 | **62.51** | 64.50 | 35.49 | **37.14** | 30.79 | 31.17 |
| | w/ SRR | **65.34**$_{\pm0.43}$ | **66.24**$_{\pm0.48}$ | **64.06**$_{\pm0.36}$ | **63.90**$_{\pm0.55}$ | 62.24$_{\pm0.13}$ | **73.19**$_{\pm0.32}$ | **37.62**$_{\pm0.79}$ | 36.53$_{\pm0.30}$ | **31.67**$_{\pm0.72}$ | **32.02**$_{\pm0.59}$ |
| **LLaMA-2 7B** | BF16 | 75.89 | | 68.51 | | 77.92 | | 40.97 | | 31.20 | |
| | *w-only* | 70.24 | | 65.67 | | 68.44 | | 28.91 | | 29.23 | |
| | QERA-exact | 72.50 | 72.63 | 66.43 | 67.32 | 73.61 | 73.39 | 33.36 | 34.12 | 29.35 | 28.94 |
| | w/ SRR | **72.88**$_{\pm0.42}$ | **72.95**$_{\pm0.71}$ | **67.09**$_{\pm0.35}$ | **68.88**$_{\pm0.58}$ | **73.72**$_{\pm0.19}$ | **74.07**$_{\pm0.21}$ | **33.57**$_{\pm0.13}$ | **35.92**$_{\pm0.46}$ | **30.98**$_{\pm0.08}$ | **30.97**$_{\pm0.53}$ |
| **LLaMA-2 13B** | BF16 | 79.37 | | 71.90 | | 80.64 | | 52.12 | | 32.57 | |
| | *w-only* | 73.33 | | 68.35 | | 71.44 | | 40.88 | | 30.88 | |
| | QERA-exact | 75.90 | 76.01 | 70.56 | 70.09 | 77.77 | 77.77 | 47.46 | 47.50 | 30.59 | 31.03 |
| | w/ SRR | **76.81**$_{\pm0.60}$ | **77.27**$_{\pm0.27}$ | **71.03**$_{\pm0.57}$ | **70.59**$_{\pm0.09}$ | **79.13**$_{\pm0.83}$ | **78.93**$_{\pm0.78}$ | **48.32**$_{\pm0.51}$ | **48.09**$_{\pm0.28}$ | **32.48**$_{\pm0.36}$ | **33.00**$_{\pm0.93}$ |
| **LLaMA-3.1 8B** | BF16 | 78.99 | | 73.09 | | 81.80 | | 63.37 | | 39.44 | |
| | *w-only* | 62.93 | | 63.14 | | 65.87 | | 34.13 | | 29.76 | |
| | QERA-exact | 71.91 | 72.61 | 70.56 | 70.24 | 68.26 | 69.72 | **49.85** | 51.02 | 30.26 | 31.67 |
| | w/ SRR | **72.82**$_{\pm0.35}$ | **73.02**$_{\pm0.49}$ | **71.06**$_{\pm0.32}$ | **70.40**$_{\pm0.69}$ | **71.79**$_{\pm0.94}$ | **76.77**$_{\pm0.74}$ | 49.51$_{\pm0.36}$ | **51.19**$_{\pm0.16}$ | **33.93**$_{\pm0.26}$ | **32.59**$_{\pm0.65}$ |
| **LLaMA-3.1 70B** | BF16 | 85.02 | | 80.03 | | 85.20 | | 75.25 | | 48.66 | |
| | *w-only* | 79.88 | | 71.90 | | 78.75 | | 65.09 | | 34.46 | |
| | QERA-exact | 81.57 | 81.61 | 77.90 | 79.16 | 81.80 | 82.84 | 69.67 | 69.90 | 41.78 | 41.87 |
| | w/ SRR | **82.47**$_{\pm0.39}$ | **82.56**$_{\pm0.24}$ | **78.11**$_{\pm0.09}$ | **79.32**$_{\pm0.14}$ | **81.98**$_{\pm0.14}$ | **83.06**$_{\pm0.05}$ | **69.80**$_{\pm0.05}$ | **70.08**$_{\pm0.03}$ | **41.83**$_{\pm0.02}$ | **41.94**$_{\pm0.03}$ |

*Table 14.* Zero-shot accuracy (↑) on five downstream benchmarks under 3-bit MXINT quantization, evaluated under rank $r = 32$ and $r = 64$. Best results are highlighted in **bold**. For SRR, results are reported as accuracy $\pm$ std over three random seeds.

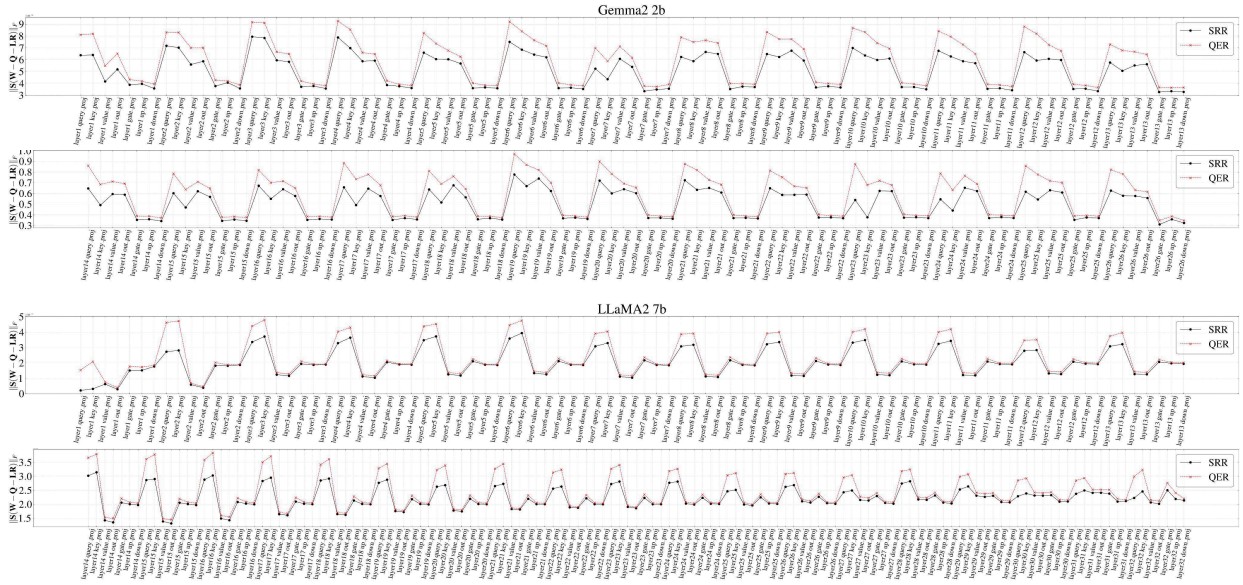

*Figure 7.* Layer-wise full reconstruction error under ZeroQuant-V2 ($\mathbf{S} = \mathbf{I}$), comparing QER and SRR. All results are obtained using 3-bit MXINT quantizer with rank 64. SRR consistently achieves lower reconstruction error than QER across all layers for both Gemma-2 2B (top) and LLaMA-2 7B (bottom).

## C.3. Scaling Behavior of SRR Across Model Sizes

To analyze the behavior of SRR across different model scales, we examine the dimension-normalized effective rank ($eRank$) of **SW** on LLaMA-2 7B and LLaMA-3.1 70B. Following prior work, we define

$$p_i = \frac{\sigma_i(\mathbf{SW})}{\sum_j \sigma_j(\mathbf{SW})}, \qquad eRank(\mathbf{SW}) = \exp\left(-\sum_i p_i \log p_i\right),$$

and report the normalized quantity $eRank(\mathbf{SW})/d$, where $d$ denotes the hidden dimension.

| Proj. | LLaMA-2 7B | LLaMA-3.1 70B |
|---|---|---|
| Key | 0.431 | 0.451 |
| Output | 0.634 | 0.639 |
| Down | 0.870 | 0.874 |

*Table 15.* Dimension-normalized effective rank ($eRank(\mathbf{SW})/d$) across model scales.

As shown in Table 15, the normalized $eRank$ remains consistent across model scales, suggesting that the spectral structure of **SW** is preserved in larger models. This indicates that the spectral properties assumed by SRR remain stable across scale.

At the same time, as model width increases, a fixed rank budget represents a progressively smaller fraction of the weight matrix. For example, with $r=32$, the low-rank component covers approximately $0.8\%$ of the hidden dimension in LLaMA-2 7B ($d=4096$), but less than $0.4\%$ in LLaMA-3.1 70B ($d=8192$). While SRR continues to optimally allocate the available rank budget, the relative correction capacity naturally decreases as the matrix dimension grows.

## C.4. Comparison with ODLRI

We clarify the distinction between ODLRI (Cho et al., 2025) and SRR. While both methods extract low-rank components prior to quantization, they address orthogonal aspects of the reconstruction process.

ODLRI focuses on how to extract low-rank components. Specifically, it uses Hessian-based input sensitivity to identify important directions for early extraction prior to quantization. In contrast, SRR focuses on how to allocate a fixed rank budget, providing a theory-guided criterion that balances structure preservation and quantization error reconstruction.

To demonstrate that this perspective provides complementary benefits, we compare SRR and ODLRI under the same QERA-exact setting using 3-bit MXINT quantizer with $r=32$.

| Method | LLaMA-2 7B | LLaMA-3.1 8B |
|---|---|---|
| ODLRI | 10.86 | 11.72 |
| SRR | **10.76** | **11.24** |

*Table 16.* Perplexity comparison between ODLRI and SRR under the same QERA-exact setting. Best results are in **bold**.

As shown in Table 16, SRR consistently achieves lower perplexity than ODLRI across both models, indicating that rank allocation plays an important role beyond the extraction strategy itself.

# D. Additional QPEFT Experiments and Analysis

**Additional Results on Gradient Scaling.** Table 17 reports the per-task GLUE results corresponding to the averaged scores in Table 6. We compare fixed attenuation factors on the preserved top-$k$ directions, $\gamma \in \{0, 0.1, 0.5, 1\}$, with SGP (Saha & Roy, 2023)'s rank-wise scaling, where gradients along the preserved directions are scaled while leaving the residual directions unscaled. Here, the two extreme settings underperform: $\gamma = 1$ does not sufficiently constrain the preserved subspace and leads to noticeable degradation, whereas $\gamma = 0$ constrains it too strongly and hampers adaptation. In contrast, moderate attenuation ($\gamma = 0.1$ and $0.5$) and SGP yield consistently strong results across GLUE tasks, reinforcing the conclusion that moderately regulating updates on the preserved directions is sufficient in practice.

| | Rank | Method | MNLI Acc. | QNLI Acc. | RTE Acc. | SST Acc. | MRPC Acc. | CoLA Matt. | QQP Acc. | STSB P/S Corr. | Avg. |
|---|---|---|---|---|---|---|---|---|---|---|---|
| **Q Bits** 4.25 | 8 | $\gamma = 0$ | 86.81 | 92.40 | 59.12 | 93.35 | 87.58 | 45.41 | 89.91 | 86.74/87.01 | 80.18 |
| | | $\gamma = 1$ | 86.26 | 92.65 | 63.31 | 93.00 | 87.78 | 53.21 | 88.88 | 88.61/88.46 | 81.70 |
| | | $\gamma = 0.5$ | 87.12 | 92.52 | 67.54 | 94.22 | 88.85 | 59.65 | 90.48 | 89.87/89.88 | 83.78 |
| | | $\gamma = 0.1$ | 87.15 | **92.67** | **72.68** | 94.27 | **89.71** | **60.07** | 90.49 | **90.00**/89.82 | **84.62** |
| | | SGP ($\alpha = 5$) | **87.18** | 92.66 | 70.95 | **94.38** | 89.66 | 60.05 | **90.52** | 89.96/**89.97** | 84.42 |
| 2.25 | 64 | $\gamma = 0$ | 82.93 | 86.26 | 53.59 | 89.72 | 70.26 | 17.90 | 87.26 | 81.56/80.22 | 71.10 |
| | | $\gamma = 1$ | 82.51 | 87.74 | 55.13 | 90.46 | 72.55 | 25.59 | 87.93 | 84.68/85.60 | 73.38 |
| | | $\gamma = 0.5$ | 83.45 | 89.68 | 57.22 | 91.63 | 86.03 | 36.16 | 89.90 | 86.70/86.36 | 77.58 |
| | | $\gamma = 0.1$ | 84.63 | 89.82 | **58.84** | 92.09 | 86.27 | 39.48 | 90.02 | 86.46/86.06 | 78.43 |
| | | SGP ($\alpha = 5$) | **85.24** | **90.43** | 58.04 | **92.43** | **87.01** | **40.71** | **90.07** | **87.23**/**86.97** | **78.88** |

*Table 17.* Full GLUE zero-shot results (↑) of SRR-based QPEFT for RoBERTa-base under MXINT quantization, comparing top-$k$ gradient scaling strategies: fixed scaling factors $\gamma \in \{0, 0.1, 0.5, 1\}$ and SGP (Saha & Roy, 2023) with $\alpha = 5$. Best results are in **bold**, with the second-best results underlined.

**SGP Hyperparameter Sensitivity.** To assess the sensitivity of SGP (Saha & Roy, 2023) to its hyperparameter, we sweep $\alpha \in \{0, 5, 10\}$ in the SGP rank-wise scaling rule (Equation 9), while keeping the quantization setting (bits) and rank fixed. Table 18 shows that performance varies only marginally across different $\alpha$ values, and the overall trend remains unchanged. SGP consistently performs comparably to the best fixed moderate scaling setting. These results indicate that QPEFT is not sensitive to the specific choice of $\alpha$ in SGP, reinforcing that the key factor is applying a soft constraint on the preserved top-$k$ directions rather than tuning the scaling rule aggressively.

| | Rank | $\alpha$ | MNLI Acc. | QNLI Acc. | RTE Acc. | SST Acc. | MRPC Acc. | CoLA Matt. | QQP Acc. | STSB P/S Corr. | Avg. |
|---|---|---|---|---|---|---|---|---|---|---|---|
| **Q Bits** 4.25 | 8 | 0 | 87.09 | 92.41 | 68.87 | 94.15 | 88.76 | 60.03 | 90.45 | 89.39/89.17 | 83.88 |
| | | 5 | **87.18** | **92.66** | **70.95** | **94.38** | **89.66** | 60.05 | **90.52** | **89.96/89.97** | **84.42** |
| | | 10 | 87.13 | 92.37 | 69.31 | 93.92 | 89.22 | **61.15** | 90.48 | 89.50/89.39 | 84.13 |
| 2.25 | 64 | 0 | 84.80 | **90.55** | 57.76 | 92.20 | 85.54 | 39.77 | 90.01 | **87.31**/87.12 | 78.48 |
| | | 5 | **85.24** | 90.43 | **58.04** | **92.43** | **87.01** | **40.71** | 90.07 | 87.23/86.97 | **78.88** |
| | | 10 | 84.30 | 90.12 | 57.40 | 92.32 | 86.03 | 40.13 | **90.22** | 87.26/**87.16** | 78.46 |

*Table 18.* SGP hyperparameter sensitivity on full GLUE results (↑) for SRR-based QPEFT under MXINT quantization. We vary the SGP exponent $\alpha \in \{0, 5, 10\}$ in the rank-wise scaling rule while keeping all other settings fixed (same quantization bits and rank). Best results are in **bold**, with the second-best results underlined.

**Applicability of SGP in QER-based QPEFT** We use SGP (Saha & Roy, 2023) in SRR-based QPEFT to balance optimization across the preserved and residual-correction terms, preventing disproportionate updates to the high-energy preserved subspace. To verify that this effect is not a generic add-on, we apply the same SGP procedure to a representative QER baseline, QERA. Unlike SRR, QERA's low-rank adapter is primarily used to absorb quantization error and thus does not induce the same preserved–correction scale separation that SGP is adopted to balance. Consequently, QERA+SGP shows no consistent gains over QERA. Moreover, in low-rank regimes (e.g., $r = 8$), SGP can further constrain the already limited adaptation subspace, effectively acting as an additional bottleneck and degrading downstream performance.

| | Rank | Method | MNLI Acc. | QNLI Acc. | RTE Acc. | SST Acc. | MRPC Acc. | CoLA Matt. | QQP Acc. | STSB P/S Corr. | Avg. |
|---|---|---|---|---|---|---|---|---|---|---|---|
| **Q Bits** 4.25 | 8 | QERA | **87.07** | **92.20** | **64.98** | **94.15** | **87.99** | **58.55** | **90.45** | **89.86/89.68** | **83.15** |
| | | QERA + SGP | 86.82 | 92.17 | 62.09 | 93.92 | 87.25 | 55.75 | 90.34 | 88.90/88.89 | 82.15 |
| 2.25 | 64 | QERA | 82.41 | **86.08** | 54.39 | **90.94** | **74.75** | **18.72** | **89.46** | **84.12**/82.50 | **72.51** |
| | | QERA + SGP | **82.55** | 85.36 | **54.51** | 88.82 | 74.26 | 12.45 | 89.35 | 83.37/**83.58** | 71.35 |

*Table 19.* Full GLUE results (↑) on RoBERTa-base under MXINT 4/2-bit quantization, comparing QERA with and without SGP. Best results are in **bold**.

# E. Empirical Validation of Assumptions

In this section, we empirically validate the assumptions introduced using LLaMA-2 7B under low-bit quantization. Unless otherwise stated, we use 3/4-bit MXINT quantizer with group size 32.

**Validation of Assumption 4.1.**    Assumption 4.1 states that, for a fixed quantizer, the scaled quantization error energy is approximately proportional to the scaled input energy, with a nearly constant proportionality factor $\eta_Q$ across layers. To evaluate this, we measure the coefficient of variation (CV), defined as $\mathrm{CV} = \sigma/\mu$, where $\sigma$ and $\mu$ denote the standard deviation and mean of $\eta_Q$ across layers, respectively. A lower CV indicates stronger consistency of the proportionality factor across layers.

**Validation of Assumption 4.2.**    Assumption 4.2 states that the spectrum of the normalized quantization error can be approximated by that of a random matrix, and is largely insensitive to the realization of the random probe matrix. To validate this assumption, we compare the true normalized quantization error spectrum $\rho_{r-k}(SE_k)$ with the one-shot random-matrix proxy $\rho_{r-k}(SE)$ across layers. We quantify their alignment using the Mean Relative Error (MRE), defined as

$$\mathbb{E}\left[\frac{|\rho_{\mathrm{act}} - \rho_{\mathrm{proxy}}|}{|\rho_{\mathrm{act}}|}\right].$$

Lower MRE indicates that the random-matrix proxy accurately captures the true quantization error spectrum.

| Quantizer | Bit | CV (Asm. 4.1) | MRE (Asm. 4.2) |
|-----------|-----|---------------|----------------|
| MXINT | 3 | 0.2112 | 0.0446 |
| MXINT | 4 | 0.1249 | 0.0231 |

*Table 20.* Empirical validation of Assumptions 4.1 and 4.2 under MXINT quantizer on LLaMA-2 7B.

**Results.**    As shown in Table 20, the CV remains moderate under 3-bit quantization and further decreases at 4-bit, supporting the layer-wise consistency of $\eta_Q$. Moreover, the MRE remains low (4.46% at 3-bit and 2.31% at 4-bit), indicating that the proposed random-matrix proxy closely matches the true quantization error spectrum.

We further evaluate both assumptions under GPTQ quantizer.

| Quantizer | CV (Asm. 4.1) | MRE (Asm. 4.2) |
|-----------|---------------|----------------|
| MXINT | 0.2112 | 0.0446 |
| GPTQ | 0.1765 | 0.1053 |

*Table 21.* Validation of Assumptions 4.1 and 4.2 under different quantizers on LLaMA-2 7B.

The CV remains moderate for both quantizers, supporting Assumption 4.1. For Assumption 4.2, MXINT exhibits a low MRE, validating the random-matrix approximation. GPTQ shows a higher MRE (approximately 10%), indicating deviation from the assumption. Consistent with prior observations on sequential Hessian-aware quantization methods, this deviation likely arises from error accumulation across channels in GPTQ (Yuan et al., 2023a), which introduces additional structure into quantization error. Nevertheless, the MRE remains low, indicating that the random-matrix approximation still captures the dominant structure of the quantization error, and SRR continues to provide consistent improvements under GPTQ.

Overall, these results empirically support both assumptions and demonstrate that the proposed approximations reliably capture the structure of quantization error in practice.

## F. Additional Training Loss Results for QPEFT Tasks

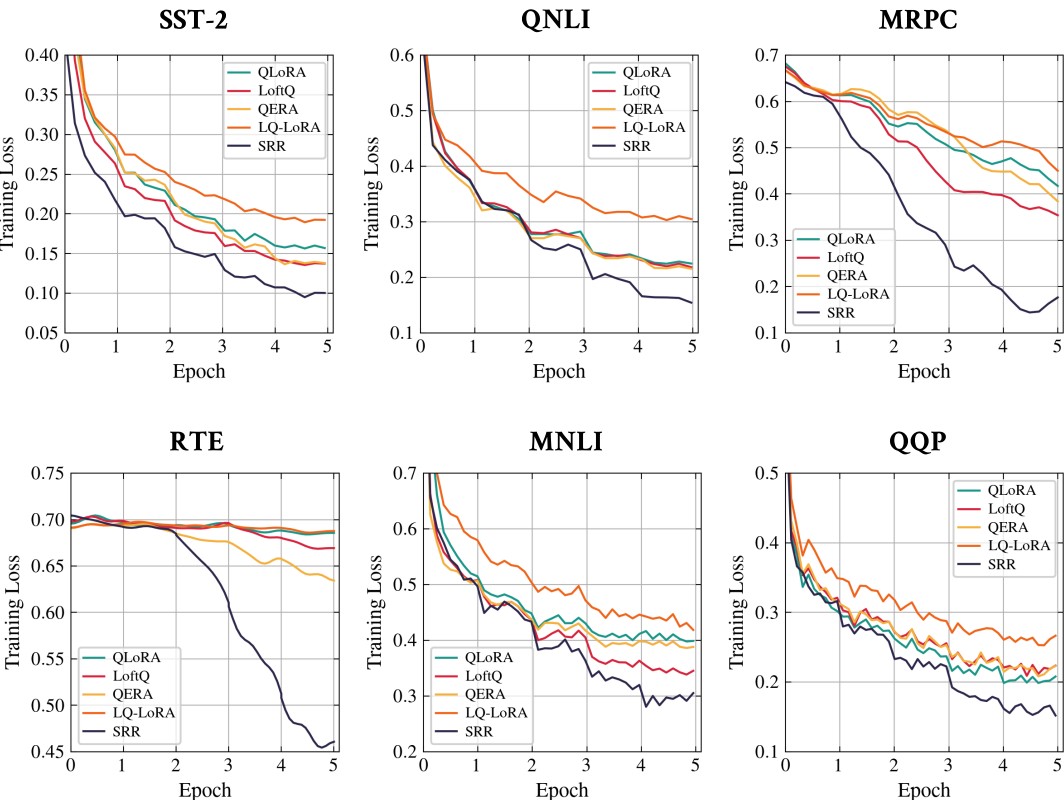

*Figure 8.* Training loss curves for QPEFT baselines on six GLUE tasks: SST-2, QNLI, MRPC, RTE, MNLI, and QQP, over 5 training epochs. Each curve corresponds to the best-performing learning rate for each method (learning rate details in Table 10). SRR tends to show a faster reduction in training loss compared to other baselines.

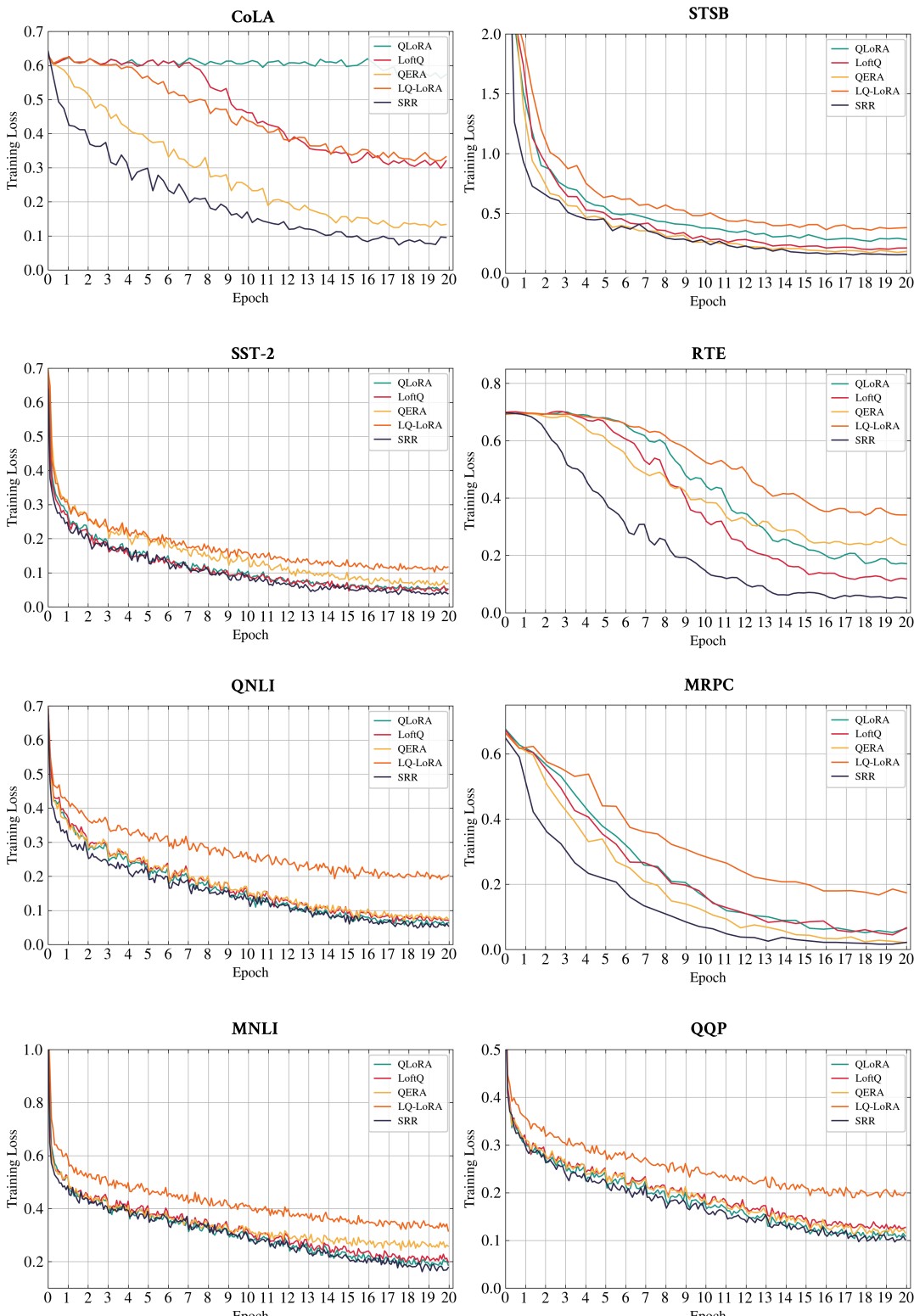

*Figure 9.* Training loss curves for QPEFT baselines on eight GLUE tasks: CoLA, STSB, SST-2, RTE, QNLI, MRPC, MNLI, and QQP, over 20 training epochs. SRR tends to show a faster reduction in training loss compared to other baselines.

