# OpenReview forum: "Preserve-Then-Quantize: Balancing Rank Budgets for Quantization Error Reconstruction in LLMs"
_ICML.cc/2026/Conference — ICML 2026 regular_

### Official Review · Reviewer_Nv4y · 2026-03-07

**Soundness:** 4
**Presentation:** 3
**Significance:** 3
**Originality:** 3
**Overall Recommendation:** 5
**Confidence:** 4

**Summary:**

This paper proposes SSR, a compression error compensation methods that applies scaled truncated SVD on weights, then quantize the residual and compensate the remaining error using scaled truncated SVD. The resultant low-rank terms can be concatenated to avoid extra matrix multiplications. SSR achieves impressive model performances compared to baseline especially when the quantization is aggressive.

**Compliance With Llm Reviewing Policy:**

Affirmed.

**Final Justification:**

The method proposed in this paper improves existing quantization error compensation method without introducing overhead.

The author addressed my concerns through rebuttal

I kept my positive score

**Key Questions For Authors:**

I'm generally positive with this paper, with the following minor questions

- Could the author offer a test of assumptions 4.1 and 4.2? I'm curious whether these assumptions hold for MXINT format
-  There are newer QER methods e.g., [SERQ](https://iclr.cc/virtual/2026/poster/10007493), which may not fully follow the QER paradigm in this paper, but a small experiments for comparison / discussion may be needed

**Limitations:**

yes

**Strengths And Weaknesses:**

## Strength
- **Clarity**: This paper is clear and easy to follow
- **Novelty**: SSR proposes new  solutions to extend exiting QER paradigm without introducing new inference-time overhead
- **Significance**: The evaluation section demonstrates a clear improvement over the QERA/LQ-LoRA baselines, especially when the quantization is aggressive, pushing QER paradigm towards 2-bits.

## Weakness

- **Test of assumptions**: Tests of assumptions for Assumption 4.1 and 4.2 may be needed as MXINT is used in this paper.
- **Recent QER baselines**: There are QER methods newer than the baselines in this paper (ZeroQuant/LQER/QERA), e.g., [SERQ](https://iclr.cc/virtual/2026/poster/10007493), but not included

---

> ### Author Rebuttal · Authors · 2026-03-31
>
> We appreciate the reviewer’s recognition of the paper’s clarity, the novelty of our work, and the clear empirical improvements over the baselines. We also thank the reviewer for the helpful suggestions, which we address below.
>
> ---
> >**Response to W1 and Q1: Test of assumptions (4.1, 4.2) in MXINT format**
>
> We sincerely thank the reviewer for the opportunity to clarify the empirical foundations of Assumptions 4.1 and 4.2. Following the reviewer's suggestion, we validated both assumptions using LLaMA-2 7B under 3-bit and 4-bit MXINT quantization (group size 32).
>
> For Assumption 4.1, which posits that the relative scale $\eta_{\mathcal{Q}}$ remains constant, we evaluated the Coefficient of Variation (CV, $σ/μ$) across all layers. A low CV justifies treating $\eta_{\mathcal{Q}}$ as a layer-independent constant. To validate Assumption 4.2, we examined the alignment between the true quantization error spectrum, $\rho_{\text{act}}$, and our one-shot random-matrix proxy, $\rho_{\text{proxy}}$. This alignment was quantified using the Mean Relative Error (MRE), $\mathbb{E}\left[ \frac{\lvert \rho_{\text{act}} - \rho_{\text{proxy}} \rvert}{\lvert \rho_{\text{act}} \rvert} \right]$, measuring the average deviation of our proxy from the ground truth.
>
> | Quantizer | Bit | CV (Asm. 4.1) | MRE (Asm. 4.2) |
> |:---:|:---:|:---:|:---:|
> | MXINT | 3 | 0.2112 | 0.0446 |
> | MXINT | 4 | 0.1249 | 0.0231 |
>
> As shown in the table above, the CV remains moderate at 3-bit and improves further to 0.1249 at 4-bit, empirically supporting the layer-wise consistency of $\eta_{\mathcal{Q}}$ required by Assumption 4.1. Furthermore, the low MRE (~4.5%) across both configurations confirms that our one-shot random-matrix proxy maintains high fidelity to the true quantization error spectrum. These results suggest that the structural assumptions of SRR are well-aligned with the empirical behavior of LLMs across different quantization settings.
> We will include these results in the revised version for completeness.
>
> ---
> >**Response to W2 and Q2: Newer QER methods (e.g., SERQ) are not included, and comparison or discussion with SERQ is needed**
>
> We thank the reviewer for bringing SERQ (Park et al., ICLR 2026) to our attention. We agree that SERQ is a highly relevant work in quantization error reconstruction, and we appreciate the opportunity to discuss this concurrent research. While both SRR and SERQ share the broader goal of error reconstruction, they are designed for fundamentally different problem settings and adopt divergent architectural strategies, which makes a direct comparison between the two methods challenging.
>
> The primary difference between SERQ and SRR lies in their targeted quantization levels and the specific error sources they address. SERQ is designed for joint weight-activation quantization (e.g., W4A4, W4A8), where activation quantization introduces additional error sources, primarily due to activation outliers. To mitigate this, SERQ incorporates activation statistics into the reconstruction process, effectively shifting part of the activation quantization-induced burden into the weight matrix.
>
> Conversely, SRR focuses on weight-only quantization in the low-bit regime while maintaining activations in high precision. Although it leverages activation statistics for scaling, SRR does not quantize activations and instead formulates the task as a global rank allocation problem over the weight matrix, balancing structure preservation and residual error reconstruction.
>
> Furthermore, the structural formulations of these methods are fundamentally distinct. SERQ focuses its reconstruction on a specific subset of salient weight rows by folding activation scales into the weights. This corresponds to a decomposition of the form $W \approx Q + R$, where the correction term $R$ is restricted to a subset of salient rows, leading to localized, row-wise compensation. Conversely, SRR adopts a $W \approx Q + LR$ parameterization, where the low-rank term spans the full weight matrix and is explicitly allocated to both preserve dominant spectral components and reconstruct residual quantization error.
>
> Given these fundamental differences in quantization settings and structural formulations, we believe a direct comparison with SERQ is not straightforward. We will incorporate a detailed clarification of these distinctions in our revised manuscript and sincerely thank the reviewer for this insightful suggestion.

---

> > ### Author Rebuttal · Reviewer_Nv4y · 2026-04-02
> >
> > The rebuttal addressed my concerns and I keep the positive score

---

### Official Review · Reviewer_7pZg · 2026-03-12

**Soundness:** 3
**Presentation:** 3
**Significance:** 3
**Originality:** 3
**Overall Recommendation:** 4
**Confidence:** 4

**Summary:**

This paper introduces Structured Residual Reconstruction (SRR), a framework for balancing the trade-off between preserving the dominant subspace of a model's weights and reconstructing quantization errors during post-training quantization (PTQ). The key contribution is a "preserve-then-quantize" approach that explicitly allocates a rank budget to first retain the top-k singular subspace of the activation-scaled weight matrix, then quantizes the residual, and finally uses the remaining rank for error reconstruction. The authors derive a theory-guided criterion for selecting the optimal k and demonstrate that SRR outperforms existing quantization error reconstruction (QER) methods in both PTQ and Quantized Parameter-Efficient Fine-Tuning (QPEFT) settings. Experiments on diverse models and tasks show consistent improvements in perplexity and task performance.

**Compliance With Llm Reviewing Policy:**

Affirmed.

**Key Questions For Authors:**

1. How sensitive is the random-matrix probe approximation to the choice of hyperparameters (e.g., probe size)? Could this affect the robustness of k selection across different models?
2. The preserved top-k subspace is critical for SRR. Is there any analysis of what semantic or functional properties this subspace captures?
3. Will the code be open source in the future?

**Limitations:**

yes

**Strengths And Weaknesses:**

Strengths:
1. The paper is well-structured, with clear motivation, methodology, and experimental analysis.
2. The paper introduces a novel "preserve-then-quantize" paradigm, distinct from prior QER methods that allocate the full rank budget to error reconstruction. The proposed SRR framework is theoretically grounded, with a principled rank-allocation strategy derived from balancing quantization-exposed energy and unrecoverable error.
3. The paper demonstrates improvements in both PTQ and QPEFT, which are widely used in deploying large language models (LLMs).

Weaknesses:
1. The theoretical derivation for selecting k relies on assumptions about the quantization-error spectrum (e.g., using a random-matrix probe). It is unclear how sensitive the method is to these approximations.

---

> ### Author Rebuttal · Authors · 2026-03-31
>
> We appreciate the reviewer’s positive assessment and helpful comments. We are encouraged by the recognition of SRR’s theoretical foundation and our novel *preserve-then-quantize* approach. We provide our detailed responses below.
>
> ---
>
> > **Response to Q1 and W1: Robustness of the Random-Matrix probe approximation**
>
> We thank the reviewer for the opportunity to clarify our experimental setup and robustness of Assumption 4.2 regarding the random-matrix proxy.
> To clarify, we set the random-matrix (probe) size to match the full dimension of the weight matrix, ensuring there are no tunable hyperparameters in the approximation.
>
> ***Validation of Assumption 4.2***
>
> To empirically validate the fidelity of our one-shot random-matrix proxy $\rho_{\text{proxy}}$, we measure its alignment with the ground-truth quantization error spectrum ($\rho_{\text{act}}$) using Mean Relative Error (MRE): $\mathbb{E}\left[ \frac{\lvert \rho_{\text{act}} - \rho_{\text{proxy}} \rvert}{\lvert \rho_{\text{act}} \rvert} \right]$.
> This evaluation is conducted across all layers of the LLaMA-2 7B under 3-bit and 4-bit MXINT quantization (group size 32).
>
> | Quantizer | Bit | MRE (Asm. 4.2) |
> |:---:|:---:|:---:|
> | MXINT | 3 | 0.0446 |
> | MXINT | 4 | 0.0231 |
>
> The resulting MRE remains low (4.46% at 3-bit and 2.31% at 4-bit), indicating that the proxy closely matches the true quantization error spectrum. This confirms that the one-shot random-matrix approximation provides a reliable estimate of the error structure in practice.
>
> ***Robustness to Random Matrix***
>
> Since the objective depends on the normalized spectral tail energy, which is highly concentrated, it is largely insensitive to the specific realization of the random probe. As detailed in Appendix B.1, we evaluated sensitivity across different random seeds and found that the selected rank $k^\*$ is highly stable, typically varying by only $\pm 1$ (and at most $\pm 3$) across layers and models.
>
> ***Performance is Insensitive to Minor Variations***
>
> Most importantly, slight variations in $k^\*$ have a negligible impact on final outcomes. As shown in Tables 1 and 5, perplexity exhibits extremely low variance across three different random seeds (each corresponding to a different random-matrix realization).
>
> Taken together, these results suggest that a single random-matrix probe provides a highly reliable and efficient estimate for rank allocation. The one-shot approximation is robust, and minor deviations in the selected $k^*$ do not compromise the model's final performance, reducing the need for expensive searches.
>
> ---
>
> > **Response to Q2: Semantic/functional properties of preserved top-k subspace**
>
> We thank the reviewer for this insightful question. To be clear, our approach does not necessarily assume an inherent semantic boundary associated with the specific value of $k$ or the top-$k$ subspace.
>
> In principle, every dimension extracted before quantization helps preserve the original structure of the activation-scaled weight matrix. Our selection of $k^\*$ is not based on isolating a functionally special class of features, but is strictly determined by a mathematical trade-off under a fixed rank budget:
> - **Pre-Quantization Benefit**: How much does extracting a specific dimension early help preserve the dominant energy of the underlying structure?
> - **Post-Quantization Benefit**: How does that benefit compare to saving that same unit of rank to compensate for the inevitable error introduced after quantization?
>
> Our algorithm simply finds the optimal mathematical break-even point between these two utilities. Importantly, this purely structural trade-off naturally aligns with the functional roles of different weight matrices. As shown in Figure 5 and Appendix B.2, Query (Q) and Key (K) projections tend to have highly concentrated spectra, meaning the early dimensions carry enough energy to justify spending the rank budget before quantization (resulting in a larger $k^*$). Conversely, Value (V) projections exhibit flatter spectra, meaning the rank budget is better spent after quantization to reconstruct the error.
>
> Therefore, while we do not explicitly analyze the semantic meaning of the subspace, the preserved dimensions reflect the intrinsic spectral properties and functional roles of each projection.
>
> ---
>
> > **Response to Q3: Clarification on public code release**
>
> Yes, we will release our code to ensure the reproducibility of the SRR framework.

---

> > ### Author Rebuttal · Reviewer_7pZg · 2026-04-03
> >
> > I thank the authors for their response. I will keep my score.

---

### Official Review · Reviewer_2L8i · 2026-03-12

**Soundness:** 3
**Presentation:** 3
**Significance:** 3
**Originality:** 3
**Overall Recommendation:** 4
**Confidence:** 4

**Summary:**

This paper identifies a previously implicit design choice in Quantization Error Reconstruction (QER) for LLMs: how to split a fixed low-rank budget r between preserving the dominant singular subspace of the activation-scaled weight matrix and reconstructing the quantization error on the residual. The proposed method, Structured Residual Reconstruction (SRR), frames this as an optimization over the split point k, with k=0 recovering standard QER methods and k=r recovering SVDQuant/LQ-LoRA. A surrogate criterion (Eq. 5) derived from two simplifying assumptions selects k* per weight matrix using only the pre-computed singular values and a single random-matrix probe, adding negligible overhead. SRR is plug-and-play on top of LQER, QERA-approx, and QERA-exact, and extends to QPEFT by using the two-component decomposition as initialization and applying gradient scaling on the preserved subspace. Experiments cover six LLMs from TinyLLaMA 1.1B to LLaMA-3.1 70B, three quantizers (MXINT, GPTQ, QuIP#), and QPEFT benchmarks including GLUE, SlimPajama, and GSM8K.

**Compliance With Llm Reviewing Policy:**

Affirmed.

**Final Justification:**

I have read the authors' rebuttal carefully and appreciate the effort put into addressing each of my concerns. The empirical validation of Assumptions 4.1 and 4.2 under both MXINT and GPTQ was exactly what I had hoped to see, and the honest acknowledgment that GPTQ deviates more from the assumptions is a welcome addition. The eRank analysis and capacity-bottleneck argument for diminishing gains at scale are reasonable explanations, and the direct comparison against ODLRI helps clarify the positioning of SRR relative to concurrent work. The QPEFT ablation (SRR + γ=1 vs. QERA + SGP) convincingly isolates the source of improvement.

That said, my overall assessment remains largely unchanged. The core contribution — exposing and optimizing the rank split between preservation and reconstruction — is clean and well-motivated, and the method is practical with negligible overhead. The QPEFT results are genuinely strong. However, the limited gains at larger scales and the approximate nature of the theoretical justification (which the authors themselves acknowledge) temper my enthusiasm somewhat. These are not fatal flaws, but they do bound the paper's impact.

I maintain my score of 4 (weak accept). The paper makes a solid, clearly presented contribution to the quantization-aware compression literature that others can build on, and the authors have been responsive and constructive throughout the discussion.

**Key Questions For Authors:**

1. How well does Assumption 4.1 hold for non-uniform quantizers such as GPTQ? Could you provide empirical measurements of η_Q variability across layers for GPTQ vs. MXINT on at least one model? If η_Q varies substantially for GPTQ, this would affect the interpretation of Table 5 results.

2. In the Eq. 6 variant (replacing Algorithm 1 Line 6 with a single rank-r SVD of S(W−Q)), are there settings where the explicit two-component split in Algorithm 1 measurably outperforms the simpler variant? If the two approaches are consistently equivalent, the simplified version may be preferable and worth presenting as the default.

3. The perplexity gains for LLaMA-2 13B and LLaMA-3.1 70B are much smaller than for 7B models. Do you have an analysis of why the benefit diminishes at scale — for instance, whether the singular value spectra of larger models are flatter, or whether quantization noise becomes more structured? A clearer account here would help readers understand the practical scope of SRR.

4. For the QPEFT setting, how does SRR initialization with standard LoRA gradient flow (γ=1) compare against QERA-exact initialization with SGP? In other words, can the QPEFT gains be attributed specifically to the two-component decomp

**Limitations:**

The paper includes a limitations section (Section 5) that briefly notes the reliance on approximate assumptions and the diminishing returns at larger scales. The discussion is somewhat brief but honest. A more detailed analysis of failure modes and the conditions under which SRR provides little benefit would be valuable, but this is not a blocking concern. I would suggest the authors expand this discussion in a revision. Overall: yes, limitations are adequately discussed.

**Strengths And Weaknesses:**

The paper's central contribution is a clean unifying formulation (Eq. 2) that exposes the rank allocation between preservation and reconstruction as an explicit, tunable design choice, rather than something implicitly fixed at k=0 or k=r as in prior work. This framing alone is valuable for conceptual clarity. The surrogate k* selection criterion is computationally lightweight — requiring only already-computed singular values and a single random-matrix probe — and Appendix B.1 demonstrates it is stable across random seeds (variation typically within ±1 rank). The method integrates with existing QER pipelines without adding inference-time memory overhead, since the two components are concatenated into a single rank-r factor. The experiments are well-controlled and cover diverse settings (six models, three quantizers, two rank budgets, both PTQ and QPEFT), with random seed reporting and consistent use of lm-eval.

The QPEFT results are the strongest empirical contribution: a 5.9 percentage-point improvement in average GLUE accuracy at 2-bit over QERA is significant and practically meaningful. The layer-wise k* analysis (Figure 5, Appendix B.2) is also informative, showing that Q/K projections tend toward higher k and V projections toward lower k, which aligns with the intuition that attention key/query interactions benefit more from spectral preservation.

There are, however, several concerns worth raising.

On soundness: Assumption 4.1 (constant relative quantization error scale η_Q) is standard for uniform quantizers but its validity for non-uniform schemes like GPTQ — which uses second-order Hessian information and per-column quantization — is not demonstrated. The paper applies SRR to GPTQ in Table 5 but does not verify whether η_Q is approximately constant across layers in that setting. Assumption 4.2, which proxies the normalized quantization error spectrum using a uniform random matrix, is an approximation justified mainly by appeal to the additive noise model. A plot comparing the actual ρ_{r-k}(SE_k) against the proxy ρ_{r-k}(SE) across representative layers would make the theoretical claims more concrete. Taken together, the theoretical framework is suggestive but not rigorous, and reviewers familiar with second-order quantization methods may find the justification thin for GPTQ/QuIP# results.

On significance: the gains diminish noticeably for larger models. For LLaMA-2 13B and LLaMA-3.1 70B, the perplexity improvements are around 0.06–0.09 in absolute terms, and for 3-bit GPTQ the reductions are marginal (e.g., 10.06→9.98 on LLaMA-2 7B with QERA-exact). The discussion does not adequately address whether this is due to flatter spectral decay in larger models, more structured quantization noise, or simply approaching a ceiling. This limits confidence in the method's value in the regimes that matter most practically.

A structural concern: Section 4.3 notes that Algorithm 1 Line 6 can be replaced by a single rank-r SVD of the total residual S(W−Q) (the Eq. 6 variant) and that it "empirically behaves like the intended split." If this simpler variant works equally well, it raises the question of whether the explicit two-component decomposition is necessary beyond the k*-dependent quantization step itself. The paper does not provide a direct comparison between Algorithm 1 and the Eq. 6 variant across diverse settings, which leaves this question open.

On the QPEFT contribution specifically: Table 6 shows that SRR initialization with γ=1 (no scaling) already scores 73.38 average GLUE, vs. QERA's 72.51. The further jump to 78.43 from gradient scaling is notable, but the paper acknowledges insensitivity to γ and reports that SGP achieves comparable results (78.88). This positions the gradient scaling as a useful but generic regularizer rather than a mechanism tightly coupled to the SRR decomposition structure.

On presentation: the paper is generally clearly written, though a few minor issues exist. The LoRA citation (Hu et al., 2022) appears in a parenthetical listing QPEFT references in Section 2, which is imprecise since LoRA itself is not QPEFT. Figure 3(a) is missing a y-axis label. The hat notation on W is used inconsistently in places. The "3.25" label in Table 1 would benefit from clearer formatting.

On positioning relative to prior work: Cho et al. (ACL Findings 2025), "Assigning Distinct Roles to Quantized and Low-Rank Matrices Toward Optimal Weight Decomposition," is cited but not experimentally compared. Given the conceptual overlap — both papers reason about how to assign roles to the quantized and low-rank components — a direct empirical comparison or at least a more substantive discussion of the differences would strengthen the related work section.

---

> ### Author Rebuttal · Authors · 2026-03-31
>
> We sincerely thank the reviewer for the valuable feedback. The comments have been helpful in improving the clarity and depth of the paper, and we will incorporate these suggestions in the revision.
>
> ---
> > **Response to Q1 & W (Soundness/Significance)**
>
> As suggested, we validate both assumptions on LLaMA-2 7B under 3-bit MXINT and GPTQ.
>
> For Asm. 4.1, we assess the constancy of $η_{\mathcal{Q}}$ via the Coefficient of Variation (CV, $σ/μ$) across layers.
>
> For Asm. 4.2, we compare the true quantization error spectrum $ρ_{\text{act}}$ with our random-matrix proxy $ρ_{\text{proxy}}$ across layers.
> We measure their alignment using Mean Relative Error (MRE), $\mathbb{E}\left[\frac{\lvertρ_{\text{act}}-ρ_{\text{proxy}}\rvert}{\lvertρ_{\text{act}}\rvert}\right]$.
>
> |Quantizer|CV(Asm. 4.1)|MRE(Asm. 4.2)|
> |-|-:|-:|
> |MXINT|0.2112|0.0446|
> |GPTQ|0.1765|0.1053|
>
> The CV remains moderate (≈0.2), supporting the constancy of $η_{\mathcal{Q}}$ across layers.
> For MRE, MXINT shows a low value (≈4%), validating the random matrix proxy, whereas GPTQ yields a higher MRE (≈10%), indicating deviation from the assumption.
> Consistent with [1], this arises from error accumulation across channels in GPTQ, explaining the weaker gains relative to MXINT.
> Nevertheless, GPTQ shows reasonably small MRE, and SRR still yields consistent gains.
>
> [1] Yuan et al., RPTQ, 2023.
>
> ---
> > **Response to Q2 & W (Structural concern)**
>
> We clarify that Eq. 6 does not remove the two-component structure but rather recombines it. Our SRR follows a strict three-step procedure:
>
> 1. Extract the leading rank-$k$ components ($L_1R_1$) from $W$.
> 2. Quantize the residual ($W-L_1R_1$) to yield the quantization error $E$.
> 3. Extract the remaining rank-$(r-k)$ components ($L_2R_2$) from $E$.
>
> By contrast, Eq. 6 shares Steps 1 and 2, but diverges in Step 3 by applying a rank-$r$ SVD directly to the combined matrix $W-Q=E+L_1R_1$. In this SVD, the top-$k$ directions still largely correspond to $L_1R_1$. The primary difference is that Eq. 6 forces the preserved rank-$k$ subspace to slightly adjust to help compensate for the quantization error. In practice, this modification yields only a marginal improvement.
>
> ---
> > **Response to Q3 and W (Significance)**
>
> We agree that perplexity gains relatively smaller at larger scales.
> To analyze this behavior, we compute the dimension-normalized effective rank (eRank) of $SW$ on LLaMA-2 7B ($d=4096$) and 13B ($d=5120$):
>
> $$p_i=\frac{σ_i(SW)}{\sum_jσ_j(SW)},\qquad eRank(SW)=\exp\left(-\sum_ip_i\log p_i\right).$$
>
> |Proj.|$eRank/d$ (7B)|$eRank/d$ (13B)|
> |-|-:|-:|
> |`K`|0.431|0.451|
> |`O`|0.634|0.639|
> |`Down`|0.870|0.874|
>
> As shown, the normalized eRank remains consistent, suggesting that spectral decay does not flatten in larger models and is not the primary cause of the diminished gains.
>
> We attribute the diminished gains primarily to a capacity bottleneck: as model size increases and weight dimensionality grows, a fixed rank budget covers a smaller fraction of the model.
> For example, $r=32$ covers 0.8% of the rank in 7B ($d=4096$), but under 0.4% in 70B ($d=8192$). While SRR optimally allocates the available budget, it remains constrained by the fundamental limit of rank capacity.
>
> A secondary factor is reduced headroom in larger models: quantized baselines (e.g., LLaMA-3.1 70B) already achieve low perplexity, leaving limited room for further improvement.
>
> ---
> > **Response to Q4 and W (QPEFT contribution)**
>
> To directly address this comparison, we compare SRR initialization with standard LoRA training ($γ=1$) against QERA-exact with SGP.
> In the 2-bit setting, SRR with $γ=1$ achieves 73.38 average GLUE, while QERA with SGP achieves 71.35 (refer to App. D, Table 17).
> This shows that SRR outperforms QERA regardless of gradient scaling.
>
> Moreover, applying SGP to QERA does not improve performance and can even degrade it (72.51 to 71.35), likely because QERA performs only error compensation, leaving no preserved directions to protect during training.
>
> These results indicate that the gains mainly stem from SRR initialization, with gradient scaling playing a secondary role.
> > **Response to W (Positioning SRR relative to prior work)**
>
> We clarify the distinction between ODLRI (Cho et al., 2025) and SRR.
> While both extract low-rank components prior to quantization, they address orthogonal aspects:
>
> - ODLRI focuses on how to extract: it uses Hessian-based input sensitivity to identify important directions for early extraction.
> - SRR focuses on rank allocation, providing a theory-guided criterion to distribute the rank budget between structure preservation and error reconstruction.
>
> To show that our perspective provides distinct gains, we compare both methods under the same QERA-exact setting (3-bit MXINT, $r=32$).
> SRR achieves lower perplexity than ODLRI on both LLaMA-2 7B (10.76 vs.10.86) and LLaMA-3.1 8B (11.24 vs.11.72).
>
> ---
> > **Response to W (Presentation clarity)**
>
> We thank the reviewer and will incorporate all suggestions in the revision.

---

> > ### Author Rebuttal · Reviewer_2L8i · 2026-04-05
> >
> > I thank the authors for the thorough and well-organized rebuttal. The empirical validation of Assumptions 4.1 and 4.2 under both MXINT and GPTQ, the clarification on the Eq. 6 variant, the eRank analysis explaining diminishing gains at scale, and the direct comparison with ODLRI collectively address my main concerns. I am satisfied with the responses and will maintain my current score.

---

### Official Review · Reviewer_4F2s · 2026-03-13

**Soundness:** 3
**Presentation:** 3
**Significance:** 2
**Originality:** 3
**Overall Recommendation:** 4
**Confidence:** 3

**Summary:**

The paper proposes a new method, coined as Structured Residual Reconstruction (SRR), to improve the quantization error reconstruction (QER) for large language models in the fine-tuning-free setting. In QER, a quantized weight matrix is obtained by a quantizer, and another low-rank matrix (e.g., $r$-rank) is determined by minimizing the quantization residual between the quantized weight matrix and the full precision weight matrix. Rather than using all of the $r$ ranks for the quantization error minimization, SRR proposes to first use $k$ ranks with $0 \le k < r$ to preserve the dominant structure pre-quantization, while leaving $r-k$ ranks for the residual minimization. SRR can be described in three steps: (i) obtain the dominant rank-$k$ structure, (ii) quantize the difference between full precision and the dominant rank-$k$ matrix, and (iii) compensate the error with the remaining $r - k$ ranks. Additional assumptions, such as constant relative scale of quantization error, are employed to approximate the calculation in these three steps. Empirical results in some comon LLMs show promising reduction of perplexity across several datasets.

**Compliance With Llm Reviewing Policy:**

Affirmed.

**Final Justification:**

This is a good paper, and hence, my recommendation.

**Key Questions For Authors:**

Could the authors elaborate further about the solution of Eq. (2) proposed in SRR (or Section 4.1)? This is properly a discussion how the proposed solution fit into the optimization objective.

**Limitations:**

Yes

**Strengths And Weaknesses:**

## Strengths
The idea of using a budge of $k$ ranks to preserve the dominant structure in the weight matrix $\mathbf{W}$ and leaving the remaining budget to minimize quantization error is intuitive, and similar the one in Principle Component Analysis. Preserving the dominant structure in the weight matrix allows a more optimal solution in finetuning-free context.

The paper also present an analysis of selecting $k$, which is the rank budget one could use to preserve the dominant structure pre-quantization.

## Weaknesses
Despite the novelty of the optimization in Eq. (2), the solution proposed in Section 4.1 is too simple and in fact, an approximation, not a true solution. In Eq. (2), it is a multi-variate optimization where both the $\Delta_{1}$, and $\Delta_{2}$ should be minimized at the same time. It is, of course, difficult to solve. And hence, SRR is proposed by first obtaining $\Delta_{1}$, then from that calculates $\Delta_{2}$. Such an approximation may be sub-optimal, which, in turn, requires certain analysis to understand how close the solution SRR produces to the optimal one.

In addition, the assumptions 4.1 and 4.2 used but are not empirically verified. Having some experiments to demonstrate that could strength the proposed method.

---

> ### Author Rebuttal · Authors · 2026-03-31
>
> We sincerely thank the reviewer for the positive feedback and insightful summary of our work. We appreciate the recognition of our rank allocation strategy and the clarity of our analysis on selecting $k$. We hope the following responses address the reviewer’s concerns.
>
> ---
> > **Response to W1 and Q1: Sequential Approximation under Assumptions 4.1 and 4.2**
>
> We appreciate the reviewer for raising this insightful point. We agree that the relationship between the joint optimization in Eq. (2) and our sequential solution in Section 4.1 should be explicitly clarified in the text.
>
> To clarify, our method is strictly a sequential optimization, rather than a joint one. However, it is fundamentally different from a naive, greedy sequence. Because direct joint optimization over $(\Delta_1, \Delta_2)$ is generally intractable for such complex non-convex objectives, most practical approaches settle for optimizing the first variable blindly, hoping for an acceptable outcome when optimizing the second.
>
> Our core contribution is transforming this into a forward-aware sequential process. By employing a look-ahead rank assignment, our method effectively "foresees" the optimal state of the second step while executing the first. We achieve this by leveraging the well-documented behaviors of quantizers and SVD (Assumptions 4.1 and 4.2, which we empirically validate below) to mathematically decouple the problem. This factorization allows us to effectively allocate the optimal rank assignment ($k^\star$) rather than searching blindly.
>
> To demonstrate this concretely, for a fixed rank $k$, let $R(\Delta_1) := S\bigl(W - \Delta_1 - \mathcal{Q}(W-\Delta_1)\bigr)$. For any feasible $(\Delta_1, \Delta_2)$, we can expand the objective function as follows:
>
> $$
> ||R(\Delta_1)-S\Delta_2||_F = \frac{||R(\Delta_1)-S\Delta_2||_F}{||R(\Delta_1)||_F}\cdot \frac{||R(\Delta_1)||_F}{||S(W-\Delta_1)||_F}\cdot\frac{||S(W-\Delta_1)||_F}{||SW||_F}\cdot ||SW||_F.
> $$
>
> This factorization reveals how our scheme foresees the objective value for a given $k$ by decoupling the components:
>
> - **Term 1**: The first factor originally couples $\Delta_1$ and $\Delta_2$. However, after optimizing over $\Delta_2$, Assumption 4.2 allows us to treat its optimal value as a spectral concentration term that is effectively independent of $\Delta_1$.
> - **Term 2**: The second factor becomes a constant under Assumption 4.1, which models the relative scale of the quantization error as proportional to the scaled input energy.
> - **Terms 3 & 4**: The third factor depends entirely on $\Delta_1$, while the fourth factor ($||SW||_F$) is a constant with respect to all optimization variables.
>
> By factorizing the objective into these independent terms, SRR does not suffer from the pitfalls of a naive sequential approximation. Instead, the factorization provides the exact mathematical foresight needed to allocate ranks effectively between the two steps, closely approximating the ideal joint optimization.
>
> We will revise Section 4.1 of the manuscript to explicitly include this narrative, clarifying both the necessity of the sequential scheme and the mechanics of our rank assignment foresight.
>
> ---
> > **Response to W2: Empirical validation on Assumptions 4.1 and 4.2**
>
> We thank the reviewer for highlighting the need for empirical validation of our Assumptions. In response, we conduct additional experiments to directly verify both assumptions using LLaMA-2 7B under 3/4-bit MXINT (group size 32).
>
> Assumption 4.1 states that, for a fixed quantizer, the scaled quantization error energy is approximately proportional to the scaled input energy, with a nearly constant proportionality factor $\eta_{\mathcal{Q}}$ across layers. To validate this, we measure the Coefficient of Variation (CV), defined as the ratio of standard deviation to mean ($\sigma/\mu$), of $\eta_{\mathcal{Q}}$ across all layers.
>
> Assumption 4.2 posits that the spectrum of the normalized quantization error can be well approximated by that of a random matrix, and is largely insensitive to the choice of $k$. To validate this, we evaluate the alignment between the true quantization error spectrum, $\rho_{\text{act}}$, and the one-shot random-matrix proxy, $\rho_{\text{proxy}}$, using the Mean Relative Error (MRE), $\mathbb{E}\left[ \frac{\lvert \rho_{\text{act}} - \rho_{\text{proxy}} \rvert}{\lvert \rho_{\text{act}} \rvert} \right]$.
>
> |Quantizer|Bit|CV (Asm. 4.1)|MRE (Asm. 4.2)|
> |:-:|:-:|:-:|:-:|
> |MXINT|3|0.2112|0.0446|
> |MXINT|4|0.1249|0.0231|
>
> As shown in the table, the CV remains moderate at 3-bit and further decreases at 4-bit, supporting the layer-wise consistency of $\eta_{\mathcal{Q}}$. Moreover, the MRE remains low (4.46% at 3-bit and 2.31% at 4-bit), indicating that the random-matrix proxy closely matches the true quantization error spectrum.
>
> Overall, these results empirically validate both assumptions and confirm that the proposed approximations reliably capture the structure of quantization error in practice.

---

> > ### Author Rebuttal · Reviewer_4F2s · 2026-04-02
> >
> > The rebuttal addresses my concerns. I am positive about the paper and will consider my recommendation after discussing with other reviewers.

---

### Decision · Program_Chairs · 2026-04-30

**Decision:**

Accept (regular)

**Comment:**

This paper proposes Structured Residual Reconstruction (SRR), a preserve-then-quantize framework that explicitly allocates a fixed low-rank budget between preserving dominant activation-scaled weight structure and reconstructing residual quantization error. Reviewers were generally positive and recommended accept or weak accept, highlighting the conceptual clarity of the method, its negligible additional overhead relative to standard QER, and the strong QPEFT results in low-bit regimes. During rebuttal, the authors clarified the sequential nature of the method, added empirical support for the main assumptions behind the rank-selection rule, explained the smaller gains at larger scales, and better distinguished the paper from closely related recent work. The main remaining issues are that the theory is still approximate, the PTQ gains are relatively modest on larger models, and the related-work discussion could more carefully position the paper against closely related low-rank-plus-quantization methods. The authors are encouraged to address these issues in the final revision.